# Deep mutational scanning reveals a correlation between degradation and toxicity of thousands of aspartoacylase variants

Martin Grønbæk-Thygesen[1], Vasileios Voutsinos [1], Kristoffer E. Johansson [1], Thea K. Schulze [1], Matteo Cagiada [1], Line Pedersen[1], Lene Clausen[1], Snehal Nariya[2], Rachel L. Powell [2], Amelie Stein [3], Douglas M. Fowler [2,4] ✉, Kresten Lindorff-Larsen [1] ✉ & Rasmus Hartmann-Petersen [1] ✉

Unstable proteins are prone to form non-native interactions with other proteins and thereby may become toxic. To mitigate this, destabilized proteins are targeted by the protein quality control network. Here we present systematic studies of the cytosolic aspartoacylase, ASPA, where variants are linked to Canavan disease, a lethal neurological disorder. We determine the abundance of 6152 of the 6260 (~98%) possible single amino acid substitutions and nonsense ASPA variants in human cells. Most low abundance variants are degraded through the ubiquitin-proteasome pathway and become toxic upon prolonged expression. The data correlates with predicted changes in thermodynamic stability, evolutionary conservation, and separate disease-linked variants from benign variants. Mapping of degradation signals (degrons) shows that these are often buried and the C-terminal region functions as a degron. The data can be used to interpret Canavan disease variants and provide insight into the relationship between protein stability, degradation and cell fitness.

The ubiquitin-proteasome system (UPS) is responsible for the majority of intracellular protein degradation and thus plays a vital role in maintaining protein homeostasis[1–3]. An important group of UPS substrates includes proteins that are thermodynamically destabilized or otherwise unable to attain their native conformation. When the cellular protein quality control (PQC) system detects such non-native proteins, chaperones, co-chaperones and specific E3 ubiquitin-protein ligases catalyze either their refolding or degradation via the UPS[3,4]. Along with physical conditions such as temperature[5], the folding and stability of a protein is determined by its amino acid sequence[6]. Accordingly, mutations that cause alterations in the amino acid sequence may affect the folding and thermodynamic stability of the protein. However, depending on the nature of the substitution and on its position in the protein structure, the effect of single amino acid substitutions may vary greatly, from increasing stability to a partial or complete destabilization of the natively folded structure, which in turn can lead to rapid degradation of the protein. Though estimates vary depending on which approach is used[7], about half of disease-causing missense variants are thought to lead to protein destabilization and degradation[7–14]. Despite recent progress in computational biology, predicting the effects of missense variants remains a significant challenge, which in turn reduces our ability to accurately identify pathogenic gene variants[15–17].

[1]Linderstrøm-Lang Centre for Protein Science, Department of Biology, University of Copenhagen, Copenhagen, Denmark. [2]Department of Genome Sciences, University of Washington, Seattle, WA, USA. [3]Department of Biology, University of Copenhagen, Copenhagen, Denmark. [4]Department of Bioengineering, University of Washington, Seattle, WA, USA. ✉e-mail: dfowler@uw.edu; lindorff@bio.ku.dk; rhpetersen@bio.ku.dk

Aspartoacylase (ASPA) (EC 3.5.1.15; UniProt entry P45381) is a 313 residue enzyme, mainly expressed in the cytosol of oligodendrocytes of the brain[18–20]. Here, it facilitates the hydrolysis of N-acetyl-aspartate (NAA), one of the most abundant amino acid-derived metabolites in the brain and a marker of brain health[21], into aspartate and acetate[22,23]. Structurally, the protein consists of a single domain with a channel leading to the active site[23]. Despite a sequence identity of only ~10% between ASPA and carboxypeptidases, their active sites share some structural similarity, likely utilizing similar catalytic mechanisms[23–25]. To our knowledge, no members of the zinc hydrolase superfamily have yet been systematically analyzed by deep mutational scanning.

Various interactions around the substrate binding cavity ensure high substrate specificity. Additionally, ASPA binds a $Zn^{2+}$ ion, which is coordinated by the conserved residues H21, E24, and H116[23,26]. Although the monomeric enzyme is active[27], several studies have demonstrated that ASPA forms homodimers[19,24,28] through a large interaction surface[23,26].

Insufficient ASPA activity caused by germline *ASPA* variants is linked to Canavan disease (CD) (OMIM: 271900), a recessive, neurodegenerative leukodystrophy, in which oligodendrocytes fail to properly myelinate neuroaxons[29]. The mechanism of pathogenicity remains somewhat obscure, with two suggested hypotheses that are not mutually exclusive: the osmotic-hydrostatic hypothesis suggests that abnormal NAA build-up in CD patients, and the resulting osmotic consequences, cause the disease[29,30]. Alternatively, according to the acetyl-lipid myelin hypothesis, insufficient acetate production from NAA, which is normally incorporated into the myelin sheath, is the underlying cause[18,30,31].

Clinically, CD patients suffer from poor muscle control, reduced cognitive capabilities, and other severe conditions. The symptoms appear within the first 3–6 months of life and worsen over time eventually leading to an early death[32–34]. Various attempts at curing CD or ameliorating the symptoms[35–39] have been reported, with gene therapy being one of the more promising[40–42]. Three clinical trials aiming at curing Canavan disease with gene therapy are currently ongoing (ClinicalTrials.gov Identifier: NCT05317780, NCT04998396, and NCT04833907). Due to the progressive nature of the disease, such an intervention would likely need to be performed early[42–44], thus making it essential to rapidly assess whether novel *ASPA* variants are pathogenic.

In a previous study, we have shown that the disease-linked ASPA C152W variant is targeted for PQC-linked degradation in both yeast and human cells[45]. To further investigate the PQC degradation of ASPA in relation to Canavan disease, we here generated a site-saturated library of *ASPA* variants and analyzed it using the variant abundance by a massively parallel sequencing (VAMP-seq) technique[46]. The resulting variant effect map comprises 6152 out of the 6260 possible single amino acid substitutions and nonsense variants (19 substitutions/residue*313 residues + 312 early stop codons + 1 wild type). This corresponds to a coverage of ~98% of all possible single-site missense and nonsense *ASPA* variants, and the results correlate with thermodynamic stability predictions and evolutionary conservation. We find that many of the low-abundance ASPA variants are toxic to the cells, indicating a tight connection between structural destabilization, PQC-linked degradation, and reduced fitness. Our data emphasize the importance of low ASPA abundance as a major mechanism of Canavan disease and reveal an intimate link between reduced thermodynamic stability, degradation, and reduced fitness.

## Results

### A massively parallel assay for ASPA protein abundance
We have previously shown that the disease-linked C152W ASPA variant is subject to chaperone-dependent proteasomal degradation[45]. To test whether this is a common trait for disease-linked ASPA variants, we applied variant abundance by massively parallel sequencing (VAMP-seq)[46] to a site-saturated and barcoded cDNA library of *ASPA* variants. In our approach, the ASPA library consists of ASPA variants fused to GFP. The library is expressed after site-specific recombination at a "landing pad" locus in human HEK293T cells (Fig. 1A). Since the plasmid does not contain a promoter, non-integrated plasmids are not expressed, while correct Bxb1-catalyzed site-specific integration at the landing pad locus leads to single-copy expression of GFP-fused ASPA. To correct for cell-to-cell variations in mRNA levels, the integrated plasmid also produces mCherry from an internal ribosomal entry site (IRES) downstream of ASPA. Fluorescence-activated cell sorting (FACS) is used to separate cells into distinct bins based on the GFP:mCherry ratio. Subsequently, the frequency of every variant in each bin is determined by sequencing the barcodes (Fig. 1A). Since integration of the plasmid at the landing pad will block expression of BFP and iCasp9[47], non-recombinant cells can be identified based on expression of BFP and depleted from the culture by adding AP1903 (Rimiducid), which specifically induces apoptosis of iCasp9 positive cells (Fig. 1A).

To test the feasibility of the assay, we initially compared wild-type (WT) ASPA and the C152W disease-linked variant[48,49], which was previously shown to be rapidly degraded via the proteasome and therefore of low abundance[45]. Indeed, fluorescence microscopy revealed a dramatically reduced abundance of the C152W variant (Fig. 1B). This was also evident by western blotting and was independent of whether GFP was fused to the N-terminus or C-terminus of ASPA (Fig. 1C). However, to allow for analyses of nonsense variants, we proceeded with the GFP fused to the N-terminus of ASPA. Flow cytometry showed that the mCherry levels were similar for WT and C152W, while the GFP level of WT was roughly 10-fold greater than that of C152W (Fig. 1D, E). Finally, since we were unable to detect endogenous ASPA in the HEK293T cells (Supplementary Fig. 1), the GFP-ASPA protein abundance is likely independent of potential heterodimer formation with endogenous WT ASPA.

### Comprehensive mapping of ASPA variant protein abundance
We generated a site-saturated library of *ASPA* missense and nonsense variants and inserted it in a frame with GFP (Fig. 1A). An oligo containing 18 random nucleotides was also inserted in the plasmid to serve as a barcode for the subsequent analyses (Fig. 1A). The resulting plasmid library was then subjected to long-read PacBio sequencing. This allowed us to match 134,176 unique barcodes with individual *ASPA* variants, which corresponds to each variant being represented on average by ~21 different barcodes, thus providing internal replicates for each variant. For all subsequent experiments, variants were identified by short-read Illumina sequencing of the barcodes.

The barcoded ASPA library was transfected into the HEK293T cell line and non-recombinant cells were eliminated with AP1903. Flow cytometry revealed that GFP:mCherry levels in the library spanned more than an order of magnitude and covered the range between the WT and C152W controls (Fig. 1D, E). FACS was used to separate the cells into four equally populated bins based on the GFP:mCherry levels (Fig. 1F). Then, Illumina sequencing of the barcodes allowed us to quantify the frequency with which each variant is found in each of the four bins (Fig. 1A, F) and calculate an abundance score ranging from 1 (WT-like abundance) to 0 (strongly reduced abundance). The average Pearson correlation between replicate experiments was 0.99 (Supplementary Fig. 2). However, we note that for the low-score variants the correlations between replicates were not as strong, indicating a poor resolution for the low-abundance variants. The final scores and standard deviations were determined based on 11 replicates and revealed the relative abundance of 5843 out of 5947 (98%) missense variants and 308 out of 312 (99%) nonsense variants (Fig. 2A).

The abundance scores were bimodally distributed, with a WT-like peak of stable variants centered on the synonymous (silent) substitutions and a peak of low-abundance variants centered on the nonsense

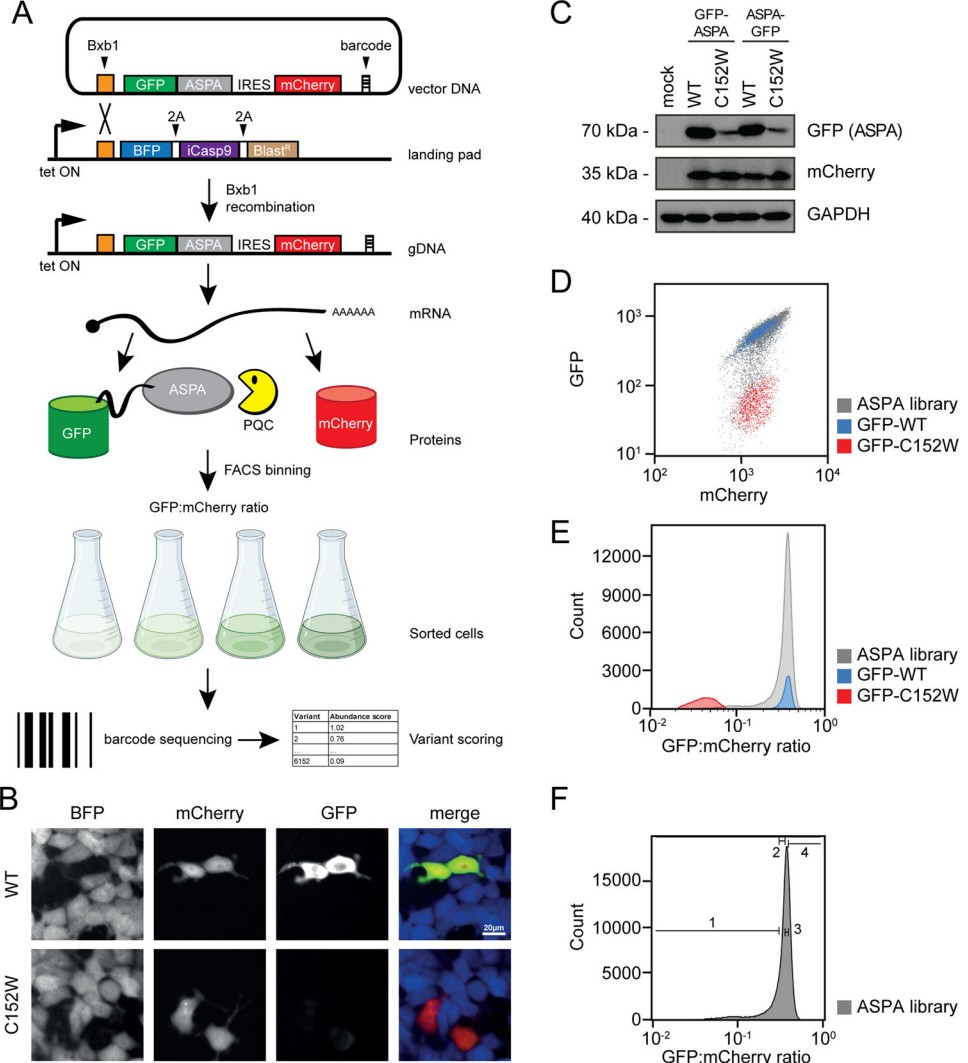

**Fig. 1 | The ASPA expression system. A** Schematic representation of the expression system. HEK293T cells, carrying a landing pad for Bxb1-catalyzed site-specific integration are transfected with the expression vector and a Bxb1 expression plasmid (not shown). Upon integration at the landing pad locus, the BFP-iCasp9-Blast$^R$ gene is displaced downstream, and the cells therefore become resistant to AP1903, while GFP-ASPA and mCherry are expressed from the tetracycline/doxycycline regulated promoter. The same mRNA leads to both GFP-ASPA and mCherry protein production, which in turn allows flow sorting of cells based on the GFP:mCherry ratio. Finally, variants in each bin can be identified by sequencing the barcodes. Figure adapted from refs. 58,76,96. Figure created with BioRender.com, released under a Creative Commons Attribution-NonCommercial-NoDerivs 4.0 International license. **B** Fluorescence microscopy of cells transfected with either wild-type ASPA (WT) or ASPA C152W variant. Note the reduced amount of the C152W variant. Scale bar = 20 µm. **C** Cells were transfected with either WT or C152W ASPA variants fused to GFP in the N-terminus or C-terminus as indicated. A mock transfection was included as a control. Whole-cell lysates were then resolved by SDS-PAGE and analyzed by Western blotting using antibodies to GFP, Cherry, or, as a loading control, GAPDH. Note the reduced level of the C152W variant. **D** Scatter plots of flow cytometry analyses of the WT (blue) and C152W (red) ASPA variants, along with the site-saturated ASPA library (gray). Note that the mCherry levels are similar, while the GFP levels differ approximately 10-fold. **E** Histograms of the GFP:mCherry ratio based on WT (blue) and C152W (red) ASPA variants, and the ASPA variant library (gray). **F** The ASPA library was sorted into four separate bins (1–4) as indicated, with each bin containing 25% of the total population.

---

variants (Fig. 2B). The abundance of 18 different ASPA variants, determined individually by flow cytometry in low throughput, was consistent with the high-throughput map (Fig. 2C). By microscopy, five variants displaying different abundance levels all appeared to localize like WT (supplementary Fig. 3A) and were largely found in the soluble fraction upon centrifugation of cell lysates (supplementary Fig. 3B). In the VAMP-seq- scores, we observe a poorer resolution of the lowest abundance variants, which appears—at least in part—to be connected to the reduced fitness observed for many of these variants (see below and discussion). The median abundance score per position (shown in Fig. 2A) explained 55% of the total variance of the variant abundance scores (equivalent to a Pearson correlation of 0.77). Thus, the tolerance to amino acid substitutions appeared more dependent on position than the nature of the substituted amino acid. However, as

expected, substitutions to proline appear detrimental at most positions (Fig. 2A). The map revealed that most regions of ASPA are sensitive to substitutions, although a particular loop stretching from position 70 to 110 appeared more tolerant (Fig. 2A), while many variants in the disordered regions (as predicted by low AlphaFold pLDDT scores) near the N- and C-termini displayed an increased abundance. In another loop spanning from position 159 to 166, substitutions to hydrophobic residues reduce ASPA abundance (Fig. 2A).

When mapping the median abundance scores at each position onto the ASPA structure, some surface regions appeared sensitive to substitutions (Fig. 2D). Most regions buried in the core of the ASPA structure were highly sensitive to substitutions (Fig. 2D), including the residues coordinating the Zn$^{2+}$ ion in the active site (Supplementary Fig. 4A). For the exposed β-strand at positions 150–160, the

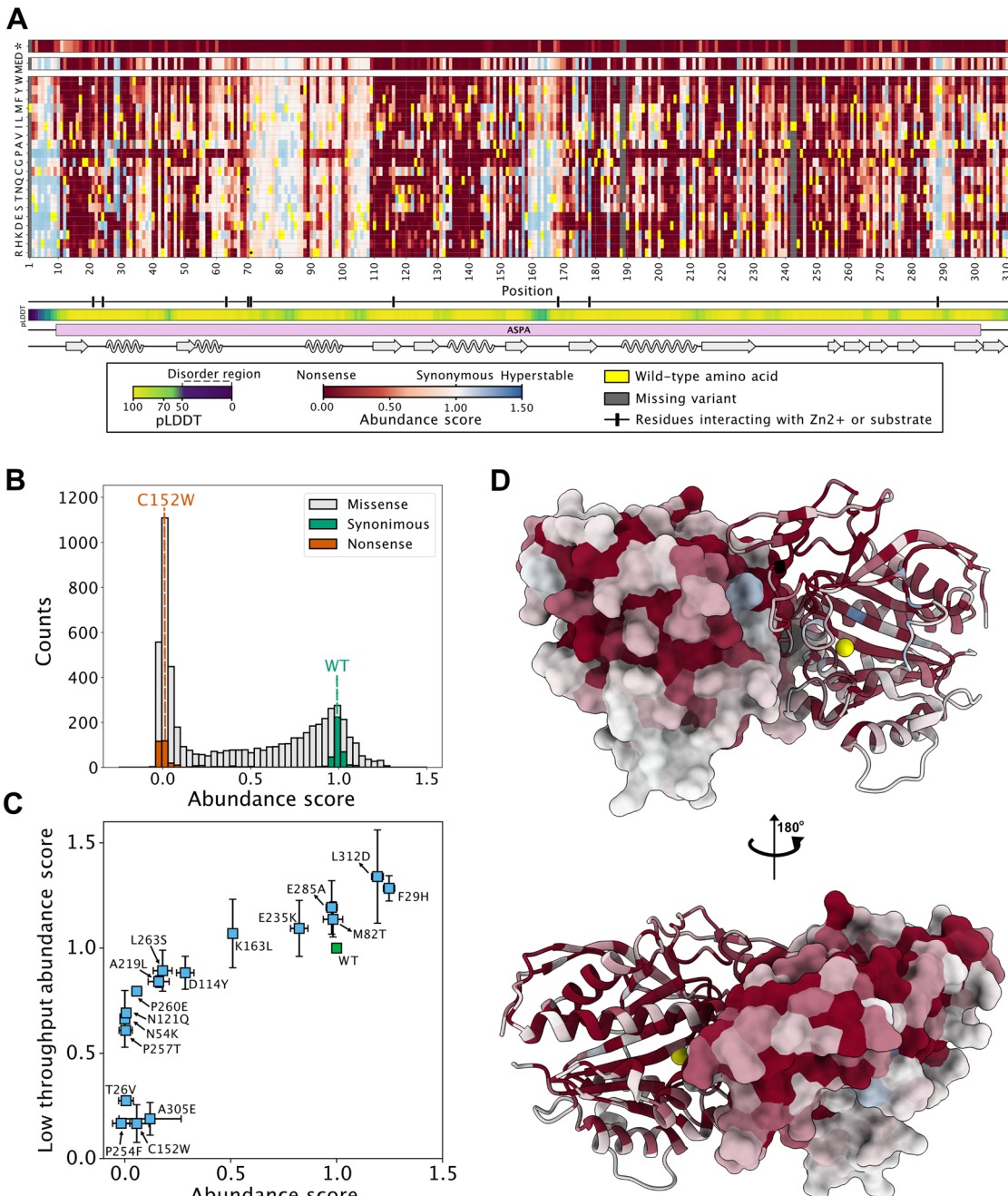

**Fig. 2 | The ASPA abundance map. A** The results of the ASPA abundance screen are presented as a heat map with the position in ASPA (horizontal) and the 20 different amino acids (vertical). * indicates a stop codon. The median abundance score (MED) per position is shown above. The wild-type residues are shown in yellow. Missing data points are marked in gray. Neutral variants (WT-like abundance) are in white. Low-abundance variants are shown in red and high-abundance variants are shown in blue. The AlphaFold confidence scores (pLDDT) are marked below, as a disorder indicator. Regions with low pLDDT scores (green/blue colors) indicate flexible/disordered regions. The ASPA domain organization and secondary structure are marked. The black bar indicates the positions of selected key catalytic residues (H21, N23, R63, N70, R71, D114, N117, E178, G185, P232, A287, Y288). **B** The library displays a bimodal distribution of abundance scores with a peak of neutral variants overlapping with the synonymous (silent) WT ASPA variants, and a peak of low-abundance variants overlapping with the nonsense ASPA variants. **C** To validate the abundance map, 18 ASPA variants were generated and analyzed one-by-one by flow cytometry in low-throughout. The abundance scores determined in low through-put (y-axis) correlate with the abundance scores determined from the screen (x-axis). The error bars reflect the standard deviation (n = 3 independent experiments). **D** The ASPA dimer structure (PDB: 2O53) colored by the median abundance score. The $Zn^{2+}$ ions are marked as yellow spheres. Note that the surface of ASPA appears more tolerant to amino acid substitutions than regions that are buried or located in the subunit-subunit interface. Source data are provided as a Source Data file.

substitutions alternated between destabilizing and stabilizing, corresponding to residues pointing inwards and outwards, respectively (Supplementary Fig. 4B). To quantify this effect more broadly, we calculated the weighted contact number[50] (WCN) for all residues in ASPA. The WCN is a sum over weights quantifying the extent to which

contacts are formed with other residues in the protein, and a high WCN thus indicates that a residue is in a densely packed region. We find that many of the low-abundance positions are buried in the structure and thus have a high WCN (Supplementary Fig. 5). However, we also find a few exposed positions with several low-

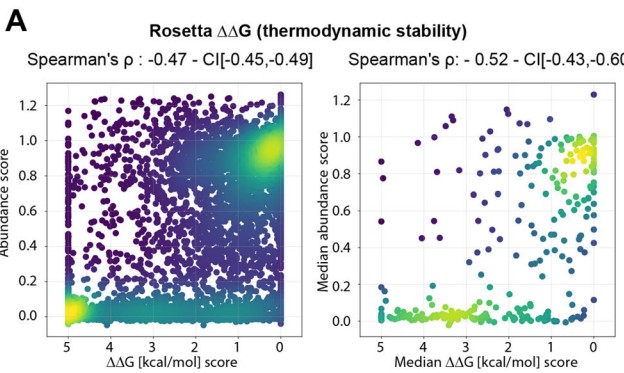

**Fig. 3 | Correlations with thermodynamic stability predictions and evolutionary conservation. A** Scatter plots showing correlations between the abundance scores and the predicted protein stabilities (ΔΔG) for all variants (left panel) and the median scores per position (right panel). **B** Scatter plots showing correlations between the abundance scores and the evolutionary conservation scores for all variants (left panel) and the median scores per position (right panel). CI, bootstrapped 95% confidence interval. Source data are provided as a Source Data file.

abundance variants, corresponding to the sensitive regions on the ASPA surface (Fig. 2D).

## The abundance of ASPA variants correlates with predicted thermodynamic folding stability

Previous reports on other proteins have suggested that variant protein abundance correlates with the experimental or predicted thermodynamic stability of the folded protein[46,51,52]. To probe this relationship for ASPA, we next employed structure-based energy calculations to predict the effects of missense variants on the thermodynamic (structural) stability of ASPA. Using the published crystal structure of the ASPA homodimer (PDB: 2O53)[26], and introducing all possible single amino acid substitutions, we applied the Rosetta energy function[53] to estimate the change (Δ) in thermodynamic folding stability (ΔG) compared to wild-type ASPA (ΔΔG). In total, the data comprise 5719 variants (19 possible amino acid substitutions per position × 301 positions resolved in the structure). The resulting ΔΔG values report on the predicted change in thermodynamic stability of the ASPA dimer, such that variants with ΔΔGs close to zero represent WT-like stability, while variants with large positive ΔΔGs should be less stable than WT ASPA, and have a higher proportion of fully or partially unfolded structures that are targeted for degradation. A comparison of the Rosetta predictions with the abundance scores represented as heat maps is included in the supplemental information (Supplementary Fig. 6A, B). Overall, the thermodynamic stability predictions correlated with the experimental abundance scores (Spearman's ρ = −0.47), which were further strengthened when comparing the median values per residue (Spearman's ρ = −0.55) (Fig. 3A). However, some variants were either predicted to be unstable (high ΔΔG) but observed at high abundance, or predicted as stable (low ΔΔG) but observed at low abundance (Fig. 3A). Hence, the thermodynamic stability predictions capture some, but not all, of the observed effects. For instance, surface-exposed sensitive regions or substitutions that introduce degrons, will not be captured. Comparisons of the abundance scores with the Rosetta predictions based on the ASPA monomer (Supplementary Fig. 6A, B) and the difference (Δ(ΔΔG)) between the Rosetta predictions for the monomer and dimer (Supplementary Fig. 6A, B), did not reveal any abundance effects that could be attributed directly to dimer formation.

## The abundance of ASPA variants correlates with evolutionary conservation

In folded proteins, residues critical for function *e.g.* those in the active site and/or for maintaining the native structure, are typically highly conserved across different species. Accordingly, sequence conservation across ASPA orthologues should predict the mutational tolerance

of the protein at the residue level. To test this, we first generated a multiple sequence alignment of 757 different ASPA homologs and then applied the GEMME[54] model that takes into account both residue conservation and the non-trivial pair couplings that occur as a consequence of amino acid co-variation. The resulting evolutionary distance scores report on the likelihood of a given substitution, where a score close to zero indicates a neutral variation with no effect on the structure and/or function of the protein. Conversely, substitutions with large negative GEMME scores are predicted as unfavorable. Again, we observed a correlation (Spearman's ρ = 0.45) between the experimental abundance scores and the predictions (Fig. 3B). As these sequence-based predictions do not discriminate between residues that are conserved for function or structure, many of the outliers in our correlations may simply be residues that are important for function but do not contribute to the thermodynamic stability of the native fold. A comparison of the GEMME predictions with the abundance scores and Rosetta stability predictions represented as heat maps is included in the supplemental information (Supplementary Fig. 7A, B).

## Most destabilized ASPA variants are heat-labile PQC and proteasome targets

Next, we proceeded to explore the cellular and physical mechanisms causing the low abundance. To this end, cells transfected with the ASPA library were subjected to a range of physical and chemical perturbations while following the distribution of the GFP:mCherry ratio of the variants by flow cytometry. The flow cytometry profiles of the WT, C152W, and the variant library in unperturbed cells are shown for comparison (Fig. 4A). First, we noted that the flow cytometry profiles for cells incubated at 29, 37, or 39.5 °C differed widely. Thus, at 39.5 °C the unstable peak became more pronounced (Fig. 4B), which suggests that at this temperature, variants with low or intermediate abundance at 37 °C are further destabilized. At 29 °C, however, the low-abundance peak was reduced to a small shoulder indicating that most variants were stabilized (Fig. 4C).

Treating the cells with an inhibitor of the E1 ubiquitin-activating enzyme (MLN7243) led to an increased abundance (Fig. 4D). Accordingly, an increased abundance was also evident when the proteasome was blocked with bortezomib (BZ) (Fig. 4E). Conversely, there was little effect of treating the cells with the autophagy-inhibitor chloroquine (CQ) (Fig. 4F). This indicates that most of the low-abundant ASPA variants are targeted by the ubiquitin-proteasome system, while autophagic clearance of ASPA variants is insignificant. However, we cannot exclude that autophagy will not play a role under conditions where the expression of the ASPA variants persists for longer.

Since HSP70-type molecular chaperones have been shown to play an important role in the PQC-linked degradation of destabilized

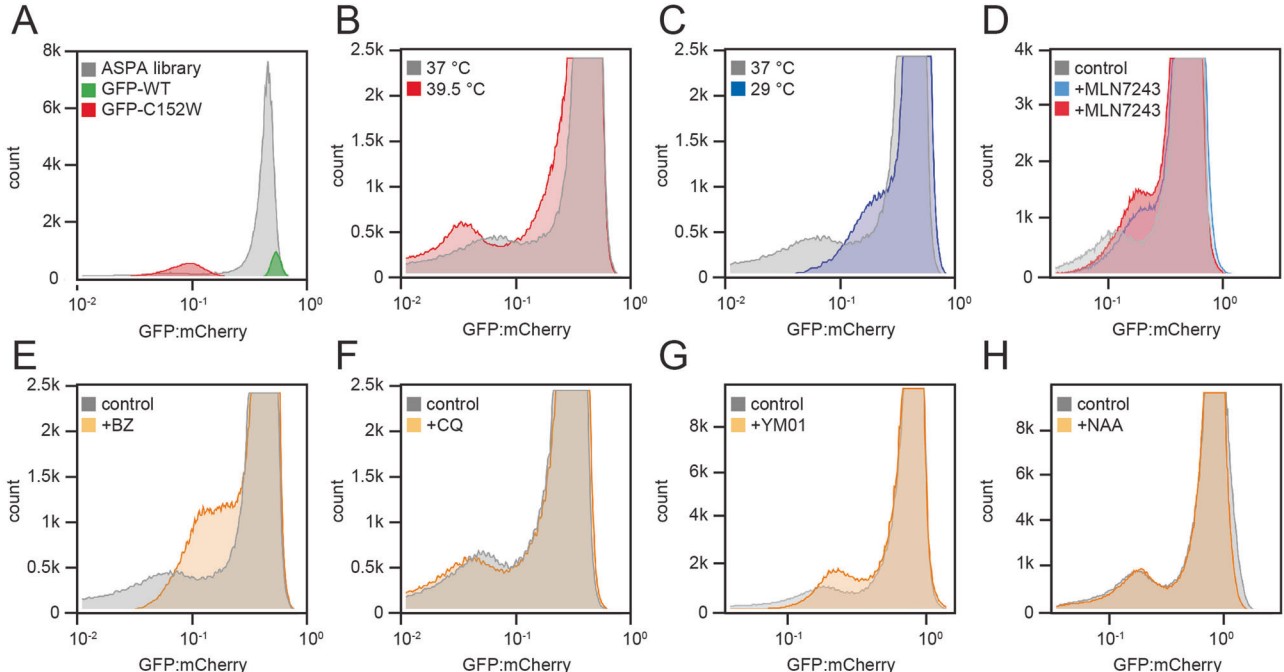

**Fig. 4 | Flow cytometry distributions of the ASPA library with different cellular perturbations.** Histograms displaying the distributions of the GFP:mCherry ratios of the ASPA library, and for comparison ASPA WT and C152W (**A**), were analyzed for the indicated perturbations: **B** 16 h incubation at 39.5 °C, **C** 16 h incubation at 29 °C, **D** 16 h at 37 °C with 0.5 µM (blue) or 1 µM (red) of the ubiquitin E1-inhibitor MLN7243, **E** 16 h at 37 °C with 15 µM of the proteasome inhibitor bortezomib (BZ), **F** 16 h at 37 °C with 20 µM the lysosomal inhibitor chloroquine (CQ), **G** 24 h at 37 °C with 2.5 µM of the HSP70 –inhibitor YM01, and **H** 24 h at 37 °C with 6 mM N-acetyl-aspartate (NAA). In all cases, perturbations were applied prior to harvesting the cells for flow cytometry.

proteins, including ASPA C152W[45], we tested the effect of the HSP70 inhibitor, YM01. Similar to the situation with bortezomib, the HSP70 inhibitor shifted the unstable peak towards a higher GFP:mCherry ratio (Fig. 4G), indicating that HSP70 plays a role in the degradation of many ASPA variants.

Since ASPA folding and stability could potentially be affected by substrate binding, we also analyzed the library distribution in the presence of NAA. However, no effects were evident upon the addition of the ASPA substrate to the cells (Fig. 4H), though we note that the resulting intracellular concentration of NAA is unknown.

Finally, as a control, we compared the abundance of WT and the C152W variant under each of the above conditions. In all cases, the abundance of WT ASPA appeared largely unaffected (supplementary Fig. 8), while–as expected–the abundance of the C152W variant was reduced at 39.5 °C but increased at 29 °C and in response to inhibition of E1, the proteasome, or HSP70 (supplementary Fig. 8).

In conclusion, these data show that most of the low-abundance ASPA variants are thermolabile targets of the ubiquitin-proteasome system. However, we cannot rule out that under different conditions, e.g. expression levels, timing, etc. some ASPA variants will undergo autophagic clearance.

## Inherent PQC degrons in ASPA map to buried regions that are sensitive to mutation

The reigning hypothesis explaining the degradation of destabilized or misfolded proteins states that these proteins, through local or global unfolding events, transiently expose PQC degradation signals (degrons). Degrons are recognized by E3 ubiquitin-protein ligases and/or molecular chaperones such as HSP70[55–58], which in turn direct the protein for proteasomal degradation.

Given the effect of inhibiting the ubiquitin-proteasome system on ASPA variants, we reasoned that ASPA likely contains PQC degrons and that mapping these degrons could shed additional light on the ASPA abundance map. To identify degrons, the ASPA sequence was divided

into 24-residue tiles each overlapping by 12 residues (Fig. 5A). Similar to full-length ASPA, we constructed a library containing the ASPA tiles that was fused to the C-terminus of GFP and expressed from the landing pad in the HEK293T cells. The cells were flow-sorted and sequencing across the tiles revealed the frequency of each ASPA tile in the four different bins (Fig. 5B). Ultimately, this allowed us to calculate a tile stability index (TSI) covering the ASPA sequence (Fig. 5C). Indeed, multiple tiles display a low TSI and thus had reduced GFP:mCherry levels, suggesting that these tiles harbor degrons. When comparing with the ASPA structure, the low-abundance tiles generally appeared buried in the structure (Supplementary Fig. 9). Accordingly, regions with low TSI also partly overlapped with regions that display a high average WCN (Fig. 5C). Comparing the mapped TSIs with PQC degron predictions made with the quality control degron predictor (QCDPred)[59,60], revealed that regions displaying a low TSI also displayed a high QCDPred degron probability, while regions with a low degron probability appeared stable (Fig. 5D). This indicates that the sequence features of the ASPA degrons are similar to those reported for PQC degrons in general, *i.e.* enriched in hydrophobic residues and depleted for acidic residues[59–62]. Finally, the C-terminal tile displayed degron properties (Fig. 5C). This may suggest that ASPA contains a C-degron[61,62] or that the C-terminal region functions as a disordered degradation initiation site (tertiary degron)[63,64], but due to the high QCDPred score could also reflect a PQC degron. We note that both substitutions and truncations in the ASPA C-terminus increase ASPA abundance (Fig. 2A), supporting the presence of a degron at this position in wild-type ASPA.

## Most disease-linked ASPA variants have reduced abundance

Next, we examined if the abundance map could distinguish known harmless (benign) and disease-linked ASPA variants. Based on the ClinVar database[65] and frequency in the population, as reported in the Genome Aggregation Database (gnomAD)[66], we first collected a curated list of disease-linked and benign ASPA variants (Source Data File).

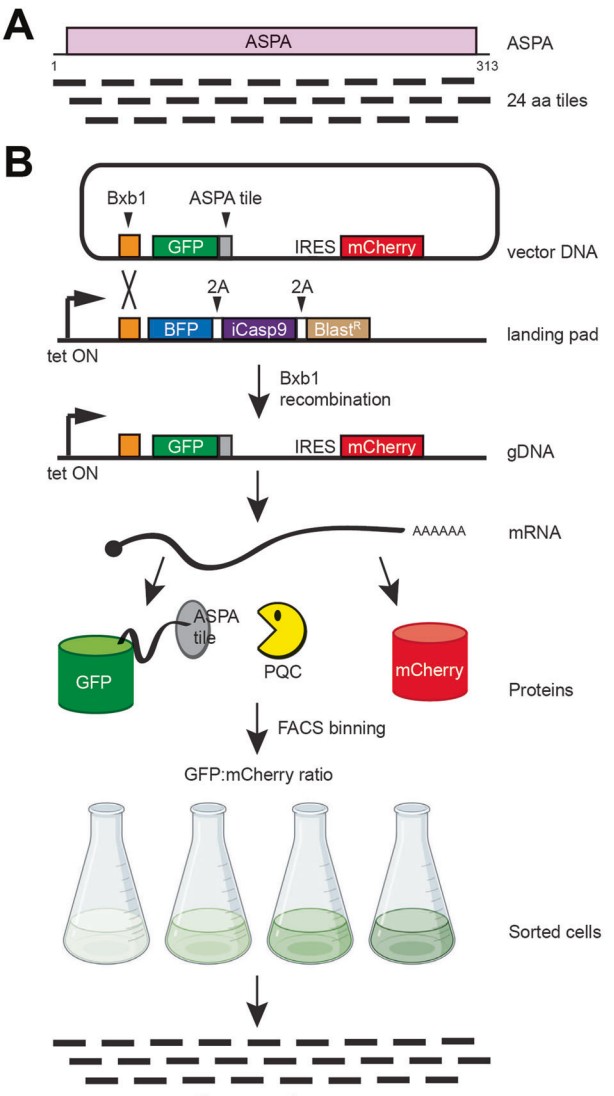

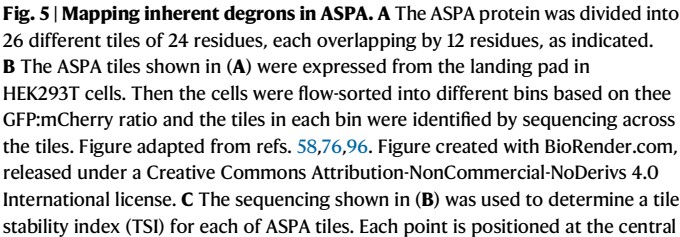

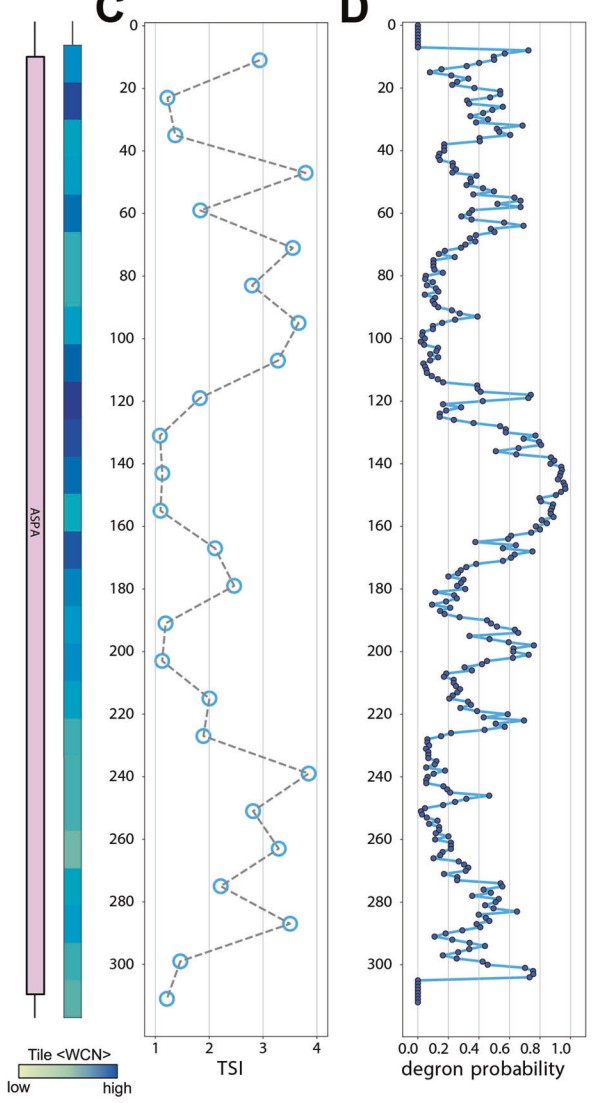

**Fig. 5 | Mapping inherent degrons in ASPA. A** The ASPA protein was divided into 26 different tiles of 24 residues, each overlapping by 12 residues, as indicated. **B** The ASPA tiles shown in (**A**) were expressed from the landing pad in HEK293T cells. Then the cells were flow-sorted into different bins based on thee GFP:mCherry ratio and the tiles in each bin were identified by sequencing across the tiles. Figure adapted from refs. 58,76,96. Figure created with BioRender.com, released under a Creative Commons Attribution-NonCommercial-NoDerivs 4.0 International license. **C** The sequencing shown in (**B**) was used to determine a tile stability index (TSI) for each of ASPA tiles. Each point is positioned at the central position of the 24-mer tiles. Tiles with a low TSI have reduced GFP:mCherry ratios and therefore display degron-like properties. As a measure for exposure, the average weighted contact number (WCN) was determined for each tile based on the ASPA crystal structure (PDB: 2O53). The domain organization of ASPA is included for comparison. **D** PQC degrons in ASPA were predicted from the ASPA sequence using QCDPred. Note that regions where QCDPred predicts a high probability of PQC degrons overlap with regions with a low TSI (**C**). Source data are provided as a Source Data file.

The three variants listed in ClinVar as benign/likely benign and also observed most frequently in the population, all displayed an abundance similar to wild-type ASPA (Fig. 6A). Conversely, 50 out of 61 pathogenic variants displayed an abundance score lower than 0.5 (Fig. 6A). Among the remaining pathogenic high-abundance variants, most were at catalytic sites (Fig. 6A, blue markers; Supplementary Fig. 10). Indeed, several of these were also predicted as stable (Rosetta $\Delta\Delta G < 2$ kcal/mol) but functionally important (GEMME score $< -3.5$) sites (Supplementary Fig. 10). Thus, these variants are likely pathogenic due to inactivating function without affecting thermodynamic stability and abundance[50]. The so-called variants of uncertain significance (VUS), *i.e.* variants where a clinical interpretation is currently lacking, clustered into high and low-abundance groups (Fig. 6A). We suggest that those with low abundance are likely to be pathogenic.

As expected, comparing the abundance scores with the allele frequencies of the ASPA variants reported in gnomAD, revealed that the most common ASPA missense alleles are benign and display wild-type-like abundance scores, while most of the low-abundance variants are rare (Fig. 6B).

Then, we examined the abundance score for the clinical variants in combination with the evolutionary conservation scores generated with GEMME (Fig. 6C). The high GEMME scores for benign and highly abundant variants indicate that these substitutions occur at evolutionary tolerant sites, indicating that they are likely functional and stable proteins. For pathogenic variants, there was a lower match with evolutionary conservation. For the 11 highly abundant pathogenic variants (abundance score > 0.5), five showed a high level of evolutionary conservation (GEMME score $< -3$), suggesting they play a role

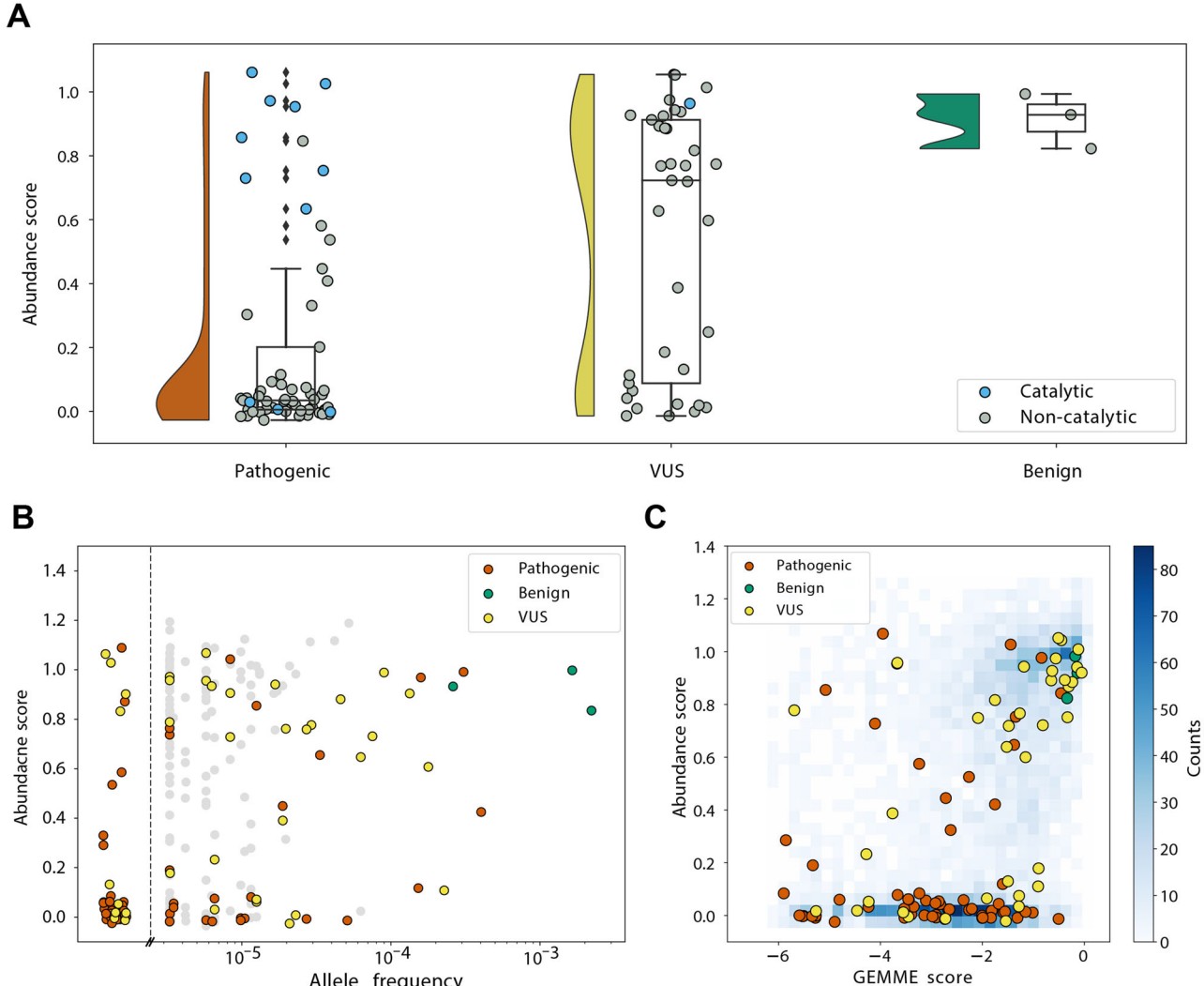

**Fig. 6 | Comparing the ASPA abundance scores with human genetics data.**
**A** Comparisons of the abundance scores for ASPA missense variants listed in the Source Data File as pathogenic (red) ($n = 61$), variants of uncertain significance (VUS) (yellow) ($n = 37$) and benign (green) ($n = 3$) are shown as raincloud plots. Residues in or near the ASPA active site have been marked (blue). Many of the high-abundance pathogenic variants are located near the active site. The box shows the quartiles of the dataset while the whiskers extend to show the rest of the distribution, except for points that are determined to be outliers (diamonds).

**B** Comparison of the ASPA abundance scores with the ASPA allele frequency reported in gnomAD. Note that ASPA variants that are common in the population are benign and display a wild-type-like abundance, while many rare variants display a low abundance. Variants so rare that they have not been observed in gnomAD are included to the left of the dashed line. **C** Comparison of the abundance scores with GEMME evolutionary conservation scores. All variants are shown as a blue 2D histogram and overlaid with variants annotated as pathogenic (red), benign (green), and VUS (yellow). Source data are provided as a Source Data file.

in enzyme activity. Additionally, most low-abundance, pathogenic variants are further distinguished from benign, abundant variants by the GEMME score.

Importantly, some ASPA variants, including G274R, P181T, Y231C, P257R, I143T, K213E, R71H, Y288C, I170T, G101V, and D204H[67–69], have been suspected to give rise to Canavan disease with a juvenile, rather than infantile onset. However, of these, all except K213E (annotated as VUS, abundance score: 0.89), R71H (annotated as pathogenic, abundance score: 0.95), and Y288C (annotated as pathogenic, abundance score: 1.1) displayed a reduced abundance, suggesting that the sensitivity of our screen may be insufficient to identify potential "mild" variants with juvenile onset.

**Certain ASPA variants become toxic upon prolonged expression**
While conducting the abundance experiments presented above, we noticed that upon prolonged incubation after inducing expression by the addition of doxycycline, cells transfected with ASPA C152W were lost from the cultures while the small group of BFP-positive cells

that survived treatment with AP1903 instead became more profuse (Supplementary Fig. 11A). Conversely, cells expressing wild-type ASPA did not disappear from cultures for at least up to 9 days (Supplementary Fig. 11B). This substantial decrease in relative growth rate suggests that prolonged expression of some non-native ASPA protein species (including C152W) is toxic to the cells. To examine this effect more broadly we therefore screened the library for effects on growth rates; in the absence of a mechanism underlying the reduction in growth we use the general term "toxicity". To this end, the library was introduced into the landing pad in the HEK293T cells, and non-recombinant cells were eliminated with AP1903. Then, without flow sorting, the cells were harvested after 0, 5, 7, and 9 days in culture and the surviving variants were identified by sequencing the barcodes (Supplementary Fig. 11C). Thus, by following the propagation of each variant in the culture over time, we could quantify the competitive fitness of each variant and calculate a "toxicity score" ranging from 1 (growth rate similar to C152W) to 0 (growth similar to WT). The final toxicity scores and standard deviations were

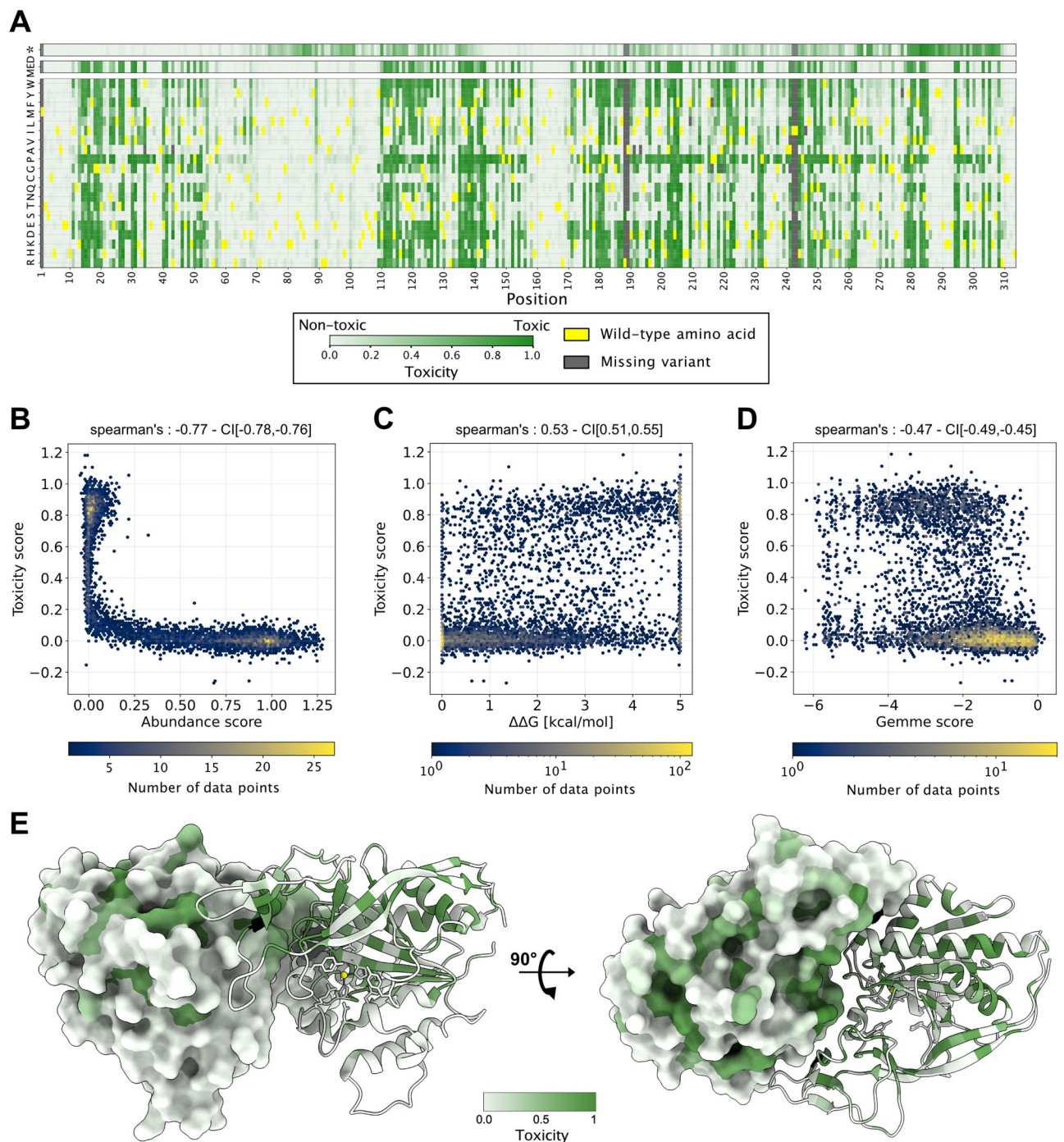

**Fig. 7 | Several low-abundant ASPA variants are toxic. A** Results from the ASPA toxicity screen presented as a heat map with the position in ASPA (horizontal) and the 20 different amino acids (vertical). * indicates a stop codon. The median toxicity score (MED) per position is shown above. The wild-type residues are shown in yellow. Missing data points are marked in gray. Non-toxic variants (WT-like) are in white. Toxic variants are shown in green. **B** Correlation between abundance and toxicity scores for all missense variants. Note that all toxic variants have low-abundance scores. **C** Plot showing the correlation between toxicity and Rosetta ΔΔG values for all missense variants. **D** Plot showing the correlation between toxicity and GEMME scores for all missense variants. In (**B**–**D**), the correlations are illustrated using 2D histograms consisting of hexagonal bins, with the number of data points in each hexagon determining the color of the bin. The data point densities are shown according to the color scales below each individual plot. CI, bootstrapped 95% confidence interval. **E** The ASPA dimer structure (PDB: 2O53) colored by the median toxicity score. The $Zn^{2+}$ ions are marked as yellow spheres. Note that toxicity is most pronounced in amino acid substitutions within regions that are buried or located in the subunit-subunit interface compared to the surface. Source data are provided as a Source Data file.

determined based on four biological replicates and the average Pearson correlation between replicate experiments was 0.93 (range: 0.93–0.94) and the mean absolute error 0.10 (Supplementary Fig. 12). Since the coverage is mainly limited by library synthesis, the coverage of toxicity scores was similar to the abundance score coverage with 5847 of 5947 (98%) missense variants and 307 out of 312 (98%) nonsense variants (Fig. 7A). The toxicity scores displayed a bimodal distribution with a peak overlapping with the synonymous WT non-toxic variants and smaller peak of toxic variants (Supplementary Fig. 13).

Comparing the toxicity scores with the abundance scores revealed that all toxic variants were low-abundance variants (Fig. 7B), indicating that continued expression of some destabilized ASPA variants is toxic, resulting in a gradual depletion of such variants from the population. The abundance levels measured in low throughput of toxic variants are lower than for non-toxic variants (Supplementary Fig. 14). This observed relation suggests that the toxicity scores also reflect cellular abundance levels. Notably, the toxicity scores are observed to resolve some of the low-abundance variants not resolved in the high-throughput abundance screen, e.g. T26V, C152W, and P254F (comparing Fig. 2C and Supplementary Fig. 14). Thus, the toxicity scores may provide a high-throughput measurement of cellular abundance levels that complements the FACS-based abundance screen in the peak around abundance score zero. Accordingly, we observe a correlation (Fig. 7C) between variant toxicity and Rosetta $\Delta\Delta G$ values (Spearman's $\rho = 0.53$), indicating that toxic variants tend to be more thermodynamically destabilized than non-toxic variants. In particular, many toxic variants are highly destabilized in our assay. However, not all variants with large positive $\Delta\Delta G$ values are toxic. Likewise, the toxic variants also appeared more unfavorable in the GEMME-based evolutionary conservation analyses (Fig. 7D). Most of the nonsense variants were non-toxic (Fig. 7A and Supplementary Fig. 13), while generally of low abundance (Fig. Fig. 2A, B). A side-by-side comparison of the toxicity, abundance, Rosetta $\Delta\Delta G$, and GEMME maps is provided in the supplemental material (Supplementary Fig. 15). Similar to the abundance map, the positions where most substitutions resulted in toxic ASPA variants were found in regions buried within the ASPA structure (Fig. 7E). Accordingly, we note a correlation (between the toxicity score and the weighted contact number (WCN)) (Supplementary Fig. 16) and a partial overlap between toxic positions and the mapped degrons (Supplementary Fig. 15B). None of the benign variants were toxic, while the pathogenic variants clustered into toxic and non-toxic groups (Supplementary Fig. 17).

Canavan disease is a recessive disorder[33], so this toxicity phenotype is unlikely to be directly relevant to the development of the disease. To examine this in more detail, we compared the gnomAD allele frequencies of the toxic and non-toxic variants. None of the gnomAD variants with a high allele frequency were toxic (Supplementary Fig. 18A). Further, the distribution of allele frequencies of low-abundance variants is similar irrespective of whether the variants were toxic or not (Supplementary Fig. 18B). Hence, while we predict that highly toxic VUS are likely to be pathogenic, the results suggest that this is due to them having low abundance rather than directly due to their toxicity in our assay.

## Toxic, low-abundance variants trigger a stress response leading to induction of HSP70

Based on the results above, we conclude that some of the thermodynamically destabilized and low-abundance ASPA variants severely reduce cellular fitness upon prolonged expression. To gain further insight into the molecular origins of this effect, we compared the transcriptomes of cells expressing the WT (non-toxic) and C152W (toxic) variants by RNA sequencing. Principal component analysis (PCA) revealed that the three independent WT samples clustered together, whereas the three C152W samples were more spread out, indicating a larger variation between the toxic samples (Supplementary Fig. 19A). Among the differentially expressed genes, we observed the small heat-shock protein HSPB8 and the HSP70-type chaperone HSPA1B as significantly upregulated in cells expressing C152W (Supplementary Fig. 19B). Among the Gene Ontology (GO) terms that were significantly enriched, we noted several related to cell stress and apoptosis (Source Data File), indicating that the toxicity is linked to a stress response caused by the degradation of thermodynamically unstable ASPA variants. Therefore, we tested if the expression of selected ASPA variants would activate the stress response pathway by

measuring the induction of the stress-responsive HSPA1B by reverse transcription qPCR. Indeed, we find a correlation between abundance, toxicity, and HSPA1B mRNA levels (Supplementary Fig. 19C–E), suggesting that the reduced growth of cells expressing certain low-abundance ASPA variants is connected with activation of the stress response pathway, which in turn inhibits cell growth.

## Discussion

Previously, we studied the protein quality control of the C152W variant in yeast cells[45], which have the advantage of being genetically tractable allowing identification of the PQC components involved in C152W folding and degradation. Although the PQC system is highly conserved across species, yeast—which does not encode any orthologue of *ASPA*—is obviously an artificial system to analyze the PQC system of relevance for Canavan disease. In the present work, we instead use human HEK293T cells. However, as these cells also do not express detectable levels of ASPA, we cannot exclude that the PQC of ASPA would be different in e.g. oligodendrocytes that appear to be the cells most relevant for Canavan disease[70].

In the present work, we probed the intimate relationship between missense protein variants, thermodynamic folding stability, degradation, and toxicity. By draining the cell for resources and through the formation of non-specific interactions, the expression of non-native proteins has long been recognized to be toxic to cells[71–73]. However, in most cases, studies on this have been limited to the expression of a single toxic protein such as Huntingtin or α-synuclein, etc.[74,75], genetically linked to dominant diseases. Since Canavan disease is a recessive disorder[33], the toxicity of certain low-abundance ASPA variants is unlikely to contribute to the disease, but rather a consequence of the variants being overexpressed, which provides us with a glimpse of how the PQC network operates. Some toxic and misfolded protein species form aggregates, and although ASPA C152W, when expressed in yeast cells, localizes to large cytosolic inclusions[45], we do not observe aggregates in the HEK293T cells. However, we cannot exclude that smaller inclusions are formed, which may influence turnover and toxicity. RNA sequencing revealed that the toxic C152W variant led to a differential expression of genes involved in stress response and apoptosis, including an upregulation of the stress-responsive HSP70-type chaperone HSPA1B. As the HSPA1B induction correlated with a reduced abundance and increased toxicity, this suggests that the toxicity is caused by the presence of destabilized ASPA variants that are prone to misfold and therefore rapidly turned over.

Most likely, the toxicity of low-abundant variants is not unique to ASPA. In a parallel study, we analyzed the abundance of a saturated library of missense variants in the protein Parkin[76]. For that protein, we did not observe any toxic effects of low-abundance variants. It is possible that the low-abundance ASPA variants are expressed at a higher level than the Parkin variants, and therefore potentially burdening the PQC system more severely. However, since Parkin is a modular protein, composed of multiple smaller domains, while ASPA is a large single-domain protein, it is also possible that ASPA unfolding events will affect the protein globally and thus be more dramatic than a putative local unfolding event localized to a single domain in Parkin. In agreement with this, we note that while most ASPA nonsense variants are of low abundance, the toxic nonsense variants primarily cluster towards the C-terminal region, indicating that the toxicity primarily occurs when the bulk of ASPA (including multiple degrons) has been produced.

Knowing that some variants can be toxic to the cells will be important for other VAMP-seq screens and other implementations of multiplex assessment of variant effects (MAVE) technologies. When considering how toxicity could impact the abundance experiment, we note the high density of variants around abundance score zero (Fig. 2B) and the elevated uncertainty in this region (Supplementary Fig. 2) may in part be a consequence of the growth effects because

toxic variants are poorly represented in terms of the number of cells. Thus, the average Spearman replica correlation of 951 non-toxic variants (toxicity score <0.4) with abundance score <0.2 is 0.77, but only 0.09 for 1529 toxic variants (toxicity score >0.6) also with abundance score <0.2. This substantial difference in replica correlations supports that the toxicity causes poor resolution in the low-abundance region also observed for the low-throughput validation experiments (Fig. 2C). Thus, although the toxicity should not affect the abundance scores directly, it may result in reduced resolution among low-abundance variants because toxicity only affects low-abundance variants and because the FACS gates are set to include the same number of cells in each bin. For future VAMP-seq analyses, sorting into more bins may increase the resolution of the low-abundance variants.

The observed correlation between protein abundance and predicted thermodynamic folding stability suggests that most low-abundance ASPA variants are thermodynamically destabilized in their structure. In turn, this will cause such variants to more frequently populate fully or partially unfolded states where regions that are buried in the native conformation become exposed. Recent structural studies on disease-linked variants in dihydrofolate reductase showed that structural destabilization led to transient exposure of a PQC degron[57]. We show that ASPA also contains multiple regions that lead to degradation when artificially exposed by grafting them as peptides onto GFP. Since most of these regions are buried in the native conformation and have sequence properties leading to high scores with QCDPred, we suggest that these fragments work as PQC degrons. Presumably, at least some of these degrons are involved in targeting non-native ASPA variants for degradation via the UPS.

Although ASPA protein abundance to some degree appears to be captured by the predicted changes in thermodynamic stability, we note several outliers in the correlation between predicted ΔΔG values and the abundance scores. Some of these could reflect experimental noise or underlying biological effects that we do not control for, e.g. introducing or destroying binding sites such as degrons or sites for post-translational modifications. We note also that Rosetta is an imperfect predictor of changes in thermodynamic stability[77], and that the relationship between thermodynamic protein stability and cellular abundance is non-linear[78,79]. We hope that new developments in the large-scale assessment of protein stability by experiments[77] and computation[10], a better understanding of quality control degradation sequences[59], and additional multiplexed measurements of abundance in eukaryotic cells will lead to improved predictors of cellular abundance.

Although we find that HSP70 contributes to the ubiquitin-dependent proteasomal turnover of many ASPA variants, we do not presently know the identity of the UPS components, including E3s, which mediate the degradation. Even though one candidate is the E3 ligase CHIP, which is known to target certain HSP70 clients for proteasomal degradation[80–82], data from yeast cells indicate a high level of redundancy between the E3s linked to the degradation of PQC substrates[83–85]. Accordingly, matching non-native proteins with their corresponding PQC E3s is not straightforward, but highly important since inhibiting such E3s should lead to increased levels of destabilized variants. In turn, this could potentially broadly alleviate genetic disorders where the variant proteins, albeit structurally destabilized, are still functional. Indeed, many disease-linked protein missense variants that are targeted for PQC-linked degradation are still, at least partially, functional[57,86,87], indicating that the PQC system is tightly tuned to root out non-native protein species. An alternative approach would be to develop small molecule stabilizers/correctors that through binding to the native conformation could block PQC-linked degradation and reactivate certain pathogenic variants. Indeed such drugs have been successfully developed and implemented for cystic fibrosis[88].

Gene therapy is currently one of the more promising attempts at curing Canavan disease[40–42,89]. However, as the disease is highly progressive, such an intervention would most likely need to be performed early[42–44]. Although MAVE assays, like those presented here, offer comprehensive genotype-phenotype information[90], they are unlikely to replace the diagnosis of Canavan disease which is currently based on elevations in N-acetyl-aspartate levels[91,92]. In addition, since pathogenic variants need not affect protein abundance, we note that high-throughput mapping of ASPA enzyme activity would likely provide a more valuable dataset from a clinical perspective, albeit on its own it would not directly inform on the molecular mechanism of pathogenicity.

## Methods

### Plasmids and library creation
The wild-type *ASPA* cDNA and selected variants studied in low throughput were generated by Genscript. The library cloning and barcoding described below are essentially as previously described[76]. The *ASPA* site-saturation mutagenesis library was purchased from Twist Biosciences and resuspended in 50 μL nuclease-free water to a final concentration of 100 ng/μL. Then, two independent 50 μL reactions with 1 μg of backbone plasmid were digested at 37 °C for 1 h with MluI-HF and EcoRI-HF (New England Biolabs). After heat inactivation (65 °C, 20 min), the products were purified following the manufacturer's protocols by resolving on a 1% agarose gel with SYBR Safe (Thermo Fisher Scientific), followed by gel extraction (Qiagen) and cleanup (Zymo Clean and Concentrate) of the 5.3 kb band. The product was assembled with the library oligonucleotide (diluted ten-fold) in a Gibson reaction (insert:backbone molar ratio of 2:1) at 50 °C for 1 h. The assembly products were then cleaned and eluted in 6 μL water (Zymo Clean and Concentrate). Then, 1 μL of Gibson assembly product was incubated with 25 μL *E. coli* NEB-10β cells (New England Biolabs) for 30 min on wet ice, prior to electroporation (2 kV, 6 ms). The cells were resuspended in 975 μL SOC (Sigma) immediately after electroporation and incubated at 37 °C for 1 h with gentle agitation. Subsequently, 1 mL culture was used to inoculate 99 mL LB media containing 100 μg/mL ampicillin and grown overnight at 37 °C. In addition, and prior to the overnight growth, to estimate library coverage by colony count, 100 μL, 10 μL, and 1 μL samples were collected and spread on LB agar plates containing 100 μg/mL ampicillin. After the overnight growth at 37 °C, the cells were harvested by centrifugation (30 min, 4300 × g) and plasmid purified by midi prep (Millipore Sigma).

### Library barcoding
For the barcoding of individual variants, 1 μg of the library plasmid was digested at 37 °C for 5 h with NdeI and SacI-HF (New England Biolabs). Then, 1 μL rSAP (New England Biolabs) was added for 30 min. at 37 °C, followed by a 20 min heat inactivation at 65 °C. The digested library was purified by 1% agarose gel electrophoresis, followed by gel extraction (Qiagen). Next, the library vectors were further purified and eluted in 10 μL water using the Zymo Clean and Concentrate kit.

The barcoding oligonucleotides, which contained 18 degenerate nucleotides, (IDT) were resuspended in water to a concentration of 10 μM. In order to anneal the barcode oligo, 1 μL of oligo was added to a mix of 1 μL 10 μM MAC356 primer, 4 μL CutSmart buffer (New England Biolabs), and 34 μL water. This reaction was incubated at 98 °C for 3 min, and then ramped down at −0.1 °C/s to 25 °C. To fill in the barcode oligo, 1.35 μL of 1 mM dNTPs and 0.8 μL Klenow exo-polymerase (New England Biolabs) were added and incubated at 25 °C for 15 min, 70 °C for 20 min, then ramped down to 37 °C at −0.1 °C/s. Once the temperature was 37 °C, the product was digested for 1 h with 1 μL each of NdeI, SacI-HF, and CutSmart buffer. Lastly, the digested product was run on a 2% agarose gel with 1× SYBR Safe (Thermo Fisher Scientific) extracted by a gel extraction kit (Qiagen) and purified further (Zymo Clean and Concentrate), followed by elution in 30 μL water.

To ligate the barcoded oligonucleotides, a ratio of 7:1 (oligo:library) was used overnight at 16 °C with T4 DNA ligase (New England

Biolabs). The products were purified and eluted in 6 μL water (Zymo Clean and Concentrate). Using the same procedure as above, *E. coli* NEB-10β cells were electroporated with 1 μL of ligation product. To bottleneck the library-barcode ligation product, electroporation recovery volumes of 500 μL, 250 μL, 125 μL, and 40 μL were separately used to inoculate 50 mL LB media containing 100 μg/mL ampicillin. Then, 100 μL, 10 μL, and 1 μL samples were spread on LB agar plates containing 100 μg/mL ampicillin, for each of the 50 mL cultures, to estimate library coverage. After growth overnight at 37 °C, the library coverage was estimated by counting colony-forming units (CFU). After overnight growth at 37 °C, each of the 50 mL cultures was centrifuged (30 min, $4300 \times g$) and plasmid purified by midi prep (Millipore Sigma). The bottlenecked library displayed an estimated 16.9 fold barcode/variant coverage.

## Subassembly of the barcode-variant map by PacBio sequencing
Using the enzymes XmaI and NdeI (New England Biolabs), 5 μg of barcoded library was digested in CutSmart buffer at 37 °C for 5 h. This was followed by heat inactivation at 65 °C for 20 min. The digested products were then purified with AMPure PB beads (Pacific Biosciences). Library preparation and DNA sequencing were performed by the University of Washington PacBio Sequencing Services. Throughout, DNA quantity was tested with fluorometry on a DS-11 FX instrument (DeNovix) with the Qubit dsDNA HS Assay Kit (Thermo Fisher Scientific). Sizes were analyzed on a 2100 Bioanalyzer (Agilent Technologies) with the High Sensitivity DNA Kit. SMRTbell sequencing libraries were generated using the protocol 'Procedure & Checklist - Preparing SMRTbell libraries using PacBio Barcoded Universal Primers for Multiplexing Amplicons' and the SMRTbell Express Template Prep Kit 2.0. The SMRTbell libraries were size-selected to remove backbone fragments using the SageELF (SageScience). The libraries were bound with Sequencing Primer v4 and Sequel II Polymerase v2.1 and sequenced on one SMRT Cell 8 M using Sequencing Plate v2.0, diffusion loading, pre-extension for 1 h, and a movie time of 30 h. Calculation of CCS consensus was performed using SMRT Link version 9.0 set at the default settings. Only reads passing an estimated quality filter of ≥Q20 were selected as "HiFi" reads.

Finally, the barcoded libraries were pooled by normalizing mass to the number of constructs contained in each pool. Then, the library was bound with Sequencing Primer v4 and Sequel II Polymerase v2.0. Sequencing was performed using SMRT Cells 8 M using Sequencing Plate v2.0, diffusion loading, a 90 min pre-extension, and a 30 h movie time. Further data were collected after SMRTbell Cleanup Kit v2 treatment to remove imperfect templates, with Sequel Polymerase v2.2. Adaptive loading with a target of 0.85 and a 1.3 h pre-extension time was used. CCS consensus and demultiplexing were performed using SMRT Link version 10.2 set at default. Reads that passed an estimated quality filter of ≥Q20 were selected for mapping barcodes to variants.

PacBio reads were filtered for reads with less than ten CSS passes using samtools version 1.16[93] and aligned to the barcode-GFP-ASPA construct using BWA version 0.7.17[94]. The barcode and ASPA sequences were extracted using cutadapt version 3.2[95], see pacbio/pacbio_align.sh available on GitHub. Reads containing ten or more DNA substitutions or any indels were filtered out. For 1,436 barcodes (1%), multiple ASPA variants mapped to the same barcode but with the majority mapping to a dominant variant, on average 89% of reads of that barcode, which was then used. This resulted in a barcode map of 134,176 unique barcodes, see pacbio/barcode_map.r on GitHub. Of these, 5970 are wild type, 6122 are synonymous wild type, and 119,301 are single amino acid variants including 5% nonsense variants. More than 98% of all possible single amino acid substitutions (incl. stop) are covered by this library and 301 of 313 positions are fully covered. Only position 1, 188, 189, 242, and 243 were missing more than 2 substitutions and the majority of these were not synthesized in the library.

Code is available at https://github.com/KULL-Centre/_2023_Groenbaek-Thygesen_ASPA_MAVE and sequencing reads at https://doi.org/10.17894/ucph.3e05fe3a-4d7e-4d70-9056-18ed999e7e1e.

## Cell propagation, transfection, and recombination
Experiments were performed using HEK293T landing pad cell line TetBxb1BFPiCasp9 Clone 12, which was characterized previously[47]. The cells were maintained in Dulbecco's Modified Eagle´s Medium (DMEM) (Sigma-Aldrich) supplemented with 10% fetal bovine serum (Sigma), 64.43 mM Penicillin G (AppliChem), 27.45 mM Streptomycin sulfate (AppliChem), and 2 mM glutamine (Sigma), with 2 μg/mL doxycycline (Dox) (Sigma-Aldrich), and split at around 80–90% confluency. For the recombination of libraries, 3.5 million cells were seeded out into a 10 cm plate with 10 mL DMEM without doxycycline. The next day, the cells were transfected as follows: In one tube, 7.1 μg library DNA and 0.48 μg pNLS-bxb1-recombinase were mixed with OptiMEM (Gibco) in a total volume of 710 μL. In another tube, 28.5 μL Fugene HD (Promega) was added to 685 μL OptiMEM and mixed gently. The two solutions were mixed gently by pipetting and incubated for 15 min before being added to the cells in 10 cm plates. Approximately 48 h later, 2 μg/ml doxycycline and 10 nM AP1903 (MedChemExpress) was added. The library was grown for 5 days in doxycycline before FACS profiling/sorting. A minimum of $10^6$ cells, corresponding to approximately 150-fold coverage of the total number of variants, were maintained in the population at all times following library recombination.

## Microscopy
Live cell fluorescence microscopy was performed using a Zeiss AxioVert microscope equipped with a digital camera (Carl Zeiss AxioCam ICm1). The following excitation and emission filters were used: BFP (excitation: $390 \pm 18$ nm, emission: $452 \pm 48$ nm), GFP (excitation: $475 \pm 28$ nm, emission: $525 \pm 48$ nm), mCherry (excitation: $575 \pm 25$ nm, emission: $620 \pm 30$ nm).

## Solubility of ASPA variants
The solubility experiments were performed as previously described[96]. Briefly, transfected cells were lysed in ice-cold buffer A (30 mM Tris/HCl, 100 mM NaCl, 5 mM EDTA, 1 mM PMSF, and complete protease inhibitor tablet (Roche), pH 7.5) followed by sonication (3 times, 20 s) on ice. The resulting whole-cell lysates were then centrifuged for 30 min at $15,000 \times g$ at 4 °C. The pellet and supernatant fractions were separated. Then, the pellet was resuspended in buffer A and the volume of this solution was set to be identical to the volume of the supernatant. Finally, SDS sample buffer was added, and the samples were analyzed by SDS-PAGE and western blotting.

## SDS-PAGE and western blotting
Cells were washed in PBS and then harvested in SDS sample buffer (3% SDS, 93 mM Tris/HCl pH 6.8, 18% glycerol, 0.02% Bromophenol blue, 2.5% (v/v) 2-mercaptoethanol) and boiled at 100 °C for 2 min. The samples were resolved on 12.5% acrylamide separation gels with a 3% (w/v) stacking gel using a constant voltage of 125 V for approximately 1 h in running buffer (50 mM Tris, 0.4 M glycine, 0.1% SDS). Next, the proteins were transferred onto a nitrocellulose membrane (pore size 0.2 μm) (Advantec), in-between filter papers (Frisenette) soaked in transfer buffer (50 mM Tris-base, 100 mM glycine, 0.01% SDS, 20% (v/v) ethanol), at 100 mAmp/gel for 1.5 h. Transferred proteins were stained in Ponceau S (0.1% Ponceau S (Sigma-Aldrich), 5% (v/v) acetic acid), Excess Ponceau was washed away with PBS (0.137 M NaCl, 2.68 mM KCl, 6.46 mM $Na_2HPO_4$, 1.47 mM $KH_2PO_4$, pH 7.4), before areas of interest were excised out of the membrane. The excised membrane pieces were incubated in blotto buffer (5% fat-free milk powder in PBS) for at least 30 min, and incubated with primary antibody overnight. Following 3 rounds of 10 min incubations in wash buffer (50 mM Tris/HCl pH 7.4, 150 mM NaCl, 0.01% (v/v) Tween-20)

the blots were incubated in horse radish peroxidase (HRP)-conjugated secondary antibodies for 1 h. After, 3 rounds of 10 min incubations in wash buffer, the blots were developed using chemiluminescence (GE Healthcare, 1059243, 1059250) on a BioRad ChemiDoc MP Imaging System imager (BioRad, 12003154). The primary antibodies used and their sources were: rabbit anti-ASPA (Thermo Fisher Scientific, PA5-29180) (diluted 1:1000), mouse anti-β-actin (Sigma-Aldrich, A5441) (diluted 1:1000), rat anti-GFP (Chromotek, 3H9) (diluted 1:1000), mouse anti-RFP (Chromotek, 6G6) (diluted: 1:1000), rabbit anti-GAPDH (Cell Signaling Technology, 14C10) (diluted 1:1000). The secondary antibodies and their sources were: HRP-anti-rat IgG (Invitrogen, 31470), HRP-anti-mouse IgG (Dako, P0260), HRP-anti-rabbit IgG (Dako, P0448), all diluted 1:5000.

## Cell sorting

Perturbations were performed as follows: For temperature, the cells were incubated at 29 °C or 39.5 °C for 16 h prior to harvesting for flow cytometry profiling. Cells (at 37 °C) were treated with 10 μM bortezomib (LC Laboratories), 20 μM chloroquine (Sigma-Aldrich), or 0.5 μM or 1 μM MLN7243 (MedChemExpress) for 16 h prior to harvesting for flow cytometry. YM01 (StressMarq Bioscience) was used at 2.5 μM and N-acetyl-aspartate (Sigma-Aldrich) at 6 mM, and added 24 h prior to harvesting for flow cytometry. For flow cytometry, cells were first washed with PBS and trypsinized (0.25% (w/v) trypsin (Gibco), 10 mM Na-citrate (Sigma-Aldrich), 102.7 mM NaCl, 0.001% phenol red (Merck, 143-74-8), pH 7.8) for 5 min at 37 °C. The dislodged cells were then washed by centrifugation in PBS and resuspended in 5% (v/v) fetal bovine serum (Sigma-Aldrich, F7524) in PBS. Then the cells were passed through a 50 μm filter (ctsv, 150–47S) into 5 mL tubes.

Cells were analyzed on a BD FACSJazz (BD Biosciences). Data was collected and analyzed using FlowJo (v10.7.2, BD), using the following gates: Live cells, singlet cells, BFP negative, and mCherry positive. We include an example of the used gating strategy (Supplementary Fig. 20).

## VAMP seq

For VAMP seq[46], cells were grown and transfected as described above. After 5 days of treatment with doxycycline, the cells were washed with PBS, trypsinized for 5 min at 37 °C, and resuspended in media. The cells were then washed by centrifugation in PBS and resuspended in 5% (v/v) fetal bovine serum in PBS. Sorting was performed with a Cell Sorter BD FACS Aria III (BD Biosciences), directly based on the GFP:mCherry ratio. In total 1.1 million cells were sorted into each of four bins. We include an example of the used gating strategy (Supplementary Fig. 20).

The cells were collected in tubes, pre-coated in 5% (v/v) fetal bovine serum in PBS overnight, and containing 1 mL media without doxycycline. Both the sample tube and the collecting tubes were kept at room temperature. After each sorting, the cells were harvested by centrifugation and resuspended in fresh media. The sorted cells were grown in 6 cm plates for 2 days (until confluent), before being resuspended and moved to 10 cm plates to grow for another 2 days (until confluent). Next, the cells were dislodged using trypsin (0.25% (w/v) trypsin (Gibco), 10 mM Na-citrate (Sigma-Aldrich), 102.7 mM NaCl, 0.001% phenol red (Merck, 143-74-8), pH 7.8) for 5 min at 37 °C, resuspended in media, before 5 million cells from each bin were isolated and centrifuged. The supernatant was aspirated and the cell pellet was stored at −80 °C for later genomic DNA extraction.

## Toxicity screen

Approximately 48 h after transfection, doxycycline and AP1903 were added to the cultures as previously described. Samples of 5 million cells were taken on days 0, 5, 7, and 9 after the introduction of doxycycline, as described for the VAMP-seq cells. For the day 0 sample,

cells were treated with doxycycline and AP1903 for 24 h to select recombinant cells, after which new media without doxycycline was added. The cells were grown until confluent and then frozen down.

## Genomic DNA extraction and sequencing

Genomic DNA extraction was performed using a Qiagen DNeasy blood & tissue kit (Cat. No. 69506). Two separate purifications were performed for each sample, to be used as technical replicates in the post-sequence analysis.

For each genomic DNA sample, an adapter PCR was performed as follows: All 8 tubes of a 50 μL PCR strip tube (VWR, catalog # 490003-606) were filled with 2500 ng DNA template, 25 μL 2× Q5 high fidelity Mastermix (New England Biolabs, M0492S), 0.5 μM forward primer (LC1020), 0.5 μM reverse primer (LC1031) and PCR-grade water to reach a total volume of 50 μL. All primers are listed in the supplemental material (Source Data File). Samples were denatured at 98 °C (30 s) followed by 7 cycles of PCR performed at temperatures: 98 °C (10 s), 60 °C (20 s), 72 °C (10 s), and lastly a final elongation step at 72 °C (2 min). The content of the PCR tubes was pooled and mixed with an equal volume of AMPure XP beads (Beckman Coulter, A63881). After 5 min incubation, the beads were pelleted and the supernatant aspirated, followed by a wash in 70% (v/v) ethanol, and elution in 21 μL PCR-grade water.

Indexing PCRs were performed by mixing: 4.1 μL PCR-grade water, 5 μL 5 μM forward primer, 5 μL 5 μM reverse primer, 25 μL 2× Q5 HF Mastermix, 2.5 μL 10× SYBR Green, 8.4 μL DNA template. Then, after initial denaturing at 98 °C (30 s), 14 PCR cycles were run at the following temperatures: 98 °C (10 s), 63 °C (20 s), 72 °C (15 s) followed by final elongation at 72 °C (2 min). Next, samples were mixed with DNA Gel Loading Dye (Thermo Fisher Scientific) and loaded onto a 2% agarose gel (2% (w/v)) agarose in TAE-buffer (40 mM Tris, 20 mM acetic acid, 1 mM EDTA with 0.01% (v/v) SYBR safe DNA gel stain (Thermo Fisher Scientific, S33102)). Electrophoresis was performed at 100 V for 50 min. Bands were visualized and excised using a Chemidoc imaging system (BioRad). The specific PCR product was extracted from the gel using the GeneJET gel extraction kit (Thermo Fisher Scientific, K0692) with a final elution in 30 μL PCR-grade water. The DNA concentration of the eluates was measured using Qubit dsDNA High Sensitivity (Thermo Fisher Scientific, Q32851) in order to normalize the pooled samples. The library was sequenced using a TG NextSeq 500/550 High Output Kit v2.5 (75 Cycles) (Illumina, 20024911). Custom primers were spiked in with the Illumina primers. Demultiplexing was performed using the BaseSpace software (Illumina).

## VAMP-seq data analyses

Illumina reads from the abundance and toxicity screens were cleaned for adapters using cutadapt[95] and paired-end reads were joined using fastq-join from ea-utils[97], see illumina/call_zerotol_paired.sh available on GitHub. Only barcodes with an exact match to the barcode map were counted, see illumina/merge_counts.r.

Read counts of barcodes were merged for amino acid variants and the technical replicates of each FACS bin (abundance) or time point (toxicity) and normalized to frequencies without pseudo counts. After merging, a score was calculated for variants with 20 or more reads observed per replica.

For the abundance scores, a protein stability index (PSI) was calculated per variant for each biological and FACS replica:

$$\text{PSI}_i = \frac{\sum_g g \times f_{i,g}}{\sum_g f_{i,g}} \quad (1)$$

where $f_{i,g}$ is the frequency of variant $i$ in FACS gate $g$. The reported scores are the mean and the standard deviation of PSI over replicates

normalized using:

$$\text{abundance score} = \frac{\text{PSI}_i - \text{PSI}_{\text{stop}}}{\text{PSI}_{\text{WT}} - \text{PSI}_{\text{stop}}} \qquad (2)$$

where $\text{PSI}_{\text{WT}}$ is the PSI value of the wild-type amino acid sequence and $\text{PSI}_{\text{stop}}$ is the median PSI value of stop substitutions per amino acid residue, both averaged over all replicates, see illumine/abundance.r available on GitHub. The average Pearson correlation of scores between replicas is 0.99 (range: 0.98–0.99) and the mean absolute error 0.05 in normalized units (Supplementary Fig. 2). All VAMP-seq data are available on GitHub and included in the supplemental material (Source Data File).

## Toxicity data analysis

The toxicity scores were calculated as the slope ($\alpha$) from a linear regression of the variant frequency at each time point. Comparable slopes are achieved by normalizing the frequencies for variant, $i$, at each time point, $t$, like a distribution of frequencies:

$$\frac{f_{i,t}}{\sum_t f_{i,t}} \qquad (3)$$

Since all sequenced pools are based on the same number of cells, we do not normalize further. This means that non-toxic variants are expected to be slightly enriched, as the complexity of the library decreases with time. We abstain from using weighted least squares because of the inherent correlation between $f_{i,t}$ and the Poisson uncertainty of this quantity which means that the expected low frequencies of toxic variants at later time points will always be down-weighted (or even ignored since pseudo counts were not applied). The reported scores are the mean and standard deviation of slopes over replicates normalized according to:

$$\text{toxicity score} = \frac{\alpha_i - \alpha_{\text{WT}}}{\alpha_{\text{C152W}} - \alpha_{\text{WT}}} \qquad (4)$$

such that wild type has a toxicity score of zero and the toxic C152W variant has a toxicity score of one, see illumine/toxicity.r available on GitHub. The toxicity data are available on GitHub and included in the supplemental material (Source Data File).

## Degron cloning

The protein sequences of seven proteins, including ASPA, were used to construct the protein-tile library presented here and in a separate manuscript[76]. The DNA sequences of these protein sequences were optimized with the IDT codon optimization tool and then split into 72 nucleotides (nt) long oligonucleotides overlapping by 36 nt except for the C-terminal tile which may have a longer overlap. To avoid unwanted PCR products produced due to template switching over the overlapping parts of the tiles, the tiles were split into odd tiles (Odds), even tiles (Evens), and C-terminal tiles (CT) based on the position they occupy in the tile series of each protein. Two 30 nt long adapters were attached to the 72 nt long sequences to serve as the complementary overlaps for Gibson assembly cloning resulting in 132 nt long oligos. Along with the 132 nt oligos three 126 nt long control oligos were made as well, each consisting of a 66 nt long oligo flanked by the same complementary Gibson overlaps. The three control oligos used were based on the APPY degron (-RLLL), which is 22 aa long sequence, and two variants that are known to mildly (-RAAA) or strongly (-DAAA) stabilize the APPY degron[58]. The 132 nt long oligos library, and the three control oligos, were ordered from IDT as three separate libraries in a way that excludes the presence of Odds, Evens, and CT oligos of the same protein in the same library tube, thus producing three

libraries referred to as Odds (complexity = 93), Evens (complexity = 91) and CT (complexity = 10).

The oligos were made into double-stranded DNA and amplified by the primers VV3 and VV4 using the following program: 98 °C for 30 s and then 98 °C for 10 s, 69 °C for 30 s 72 °C for 10 s for 2 cycles in total, followed by a final 72 °C incubation for 2 min. The PCR product was run on a 2% agarose gel with 1× SYBR Safe (Thermo Fisher Scientific) and the PCR product band was extracted with the GeneJet gel extraction kit (Thermo Fisher Scientific).

The attB-EGFP-PTEN-IRESmCherry_562Bgl[46] vector backbone was linearized by inverse PCR with primers VV1 and VV2. The reaction was run with 5 ng of the vector DNA as a template with the following program: 98 °C for 30 s and then 98 °C for 5 s, 69 °C for 30 s 72 °C for 3 min, and 40 s for 30 cycles in total, followed by a final 72 °C incubation for 5 min. The PCR product was cleaned and concentrated with the Zymo Research kit following the manufacturer's protocol and then digested by DpnI (New England BioLabs) overnight. The digestion reaction product was run on a 1% agarose gel with 1x SYBR Safe (Thermo Fisher Scientific) and the digested band was extracted from the gel with the GeneJet gel extraction kit (Thermo Fisher Scientific).

The double-stranded oligos from all three libraries (Odds, Evens, and CT) were assembled into the attB-EGFP-PTEN-IRESmCherry_562Bgl linearized vector by performing Gibson reaction mixing the oligos with the vector in a 4:1 molar ratio. The Gibson reaction was then cleaned and concentrated with the Zymo Research kit and transformed by electroporation into NEB-10β electro-competent *E. coli* cells with 2 kV. The electroporated cells were incubated for 1 h at 37 °C in 1 mL and then 100 μL of a 100-fold dilution was plated on LB-ampicillin plates. The rest (900 μL) of the transformed cells were inoculated in 100 mL LB-ampicillin liquid cultures and incubated overnight. After making sure that the CFUs on the plates were at least 100× of the complexity of each library, plasmid DNA was extracted from 100 mL cultures using a midi prep kit (Millipore Sigma), and the DNA concentration was determined by NanoDrop spectrometer ND-1000.

## Tile scoring

The tiles were integrated into the HEK 293 T TetBxb1BFPiCasp9 Clone 12 cell line as full-length ASPA, and sorted into 4 bins based on their GFP:mCherry ratio. DNA was extracted from the bins and amplicons were prepared for downstream Illumina high-throughput sequencing. Amplicons were amplified with primers VV40S and VV2S. The program of the first PCR (adapter PCR) was the following: initial denaturation was performed at 98 °C for 30 s; followed by 7 cycles of denaturation at 98 °C for 10 s, annealing at 65.5 °C for 10 s, and extension at 72 °C for 50 s; a final extension at 72 °C for 2 min. Afterwards, the product was purified by Ampure XP beads (Beckman Coulter) (0.8:1 ratio) (beads: PCR product) and the Illumina cluster generation sequences were added with a second PCR (indexing PCR) with the primers gDNA_2nd and JS_R. The PCR program used for the second PCR is as follows: initial denaturation at 98 °C for 30 s; followed by 16 cycles of denaturation at 98 °C for 10 s, annealing at 63.5 °C for 10 s, and extension at 72 °C for 10 s. The amplicons were sequenced by a NextSeq 550 sequencer with a NextSeq 500/550 Mid Output v2.5 300 cycle kit (Illumina) with custom sequencing primers VV16 and VV18 for read 1 and read 2 (paired-end). The indices were read with the primers VV19 and VV21 for index 1 and index 2 respectively.

Similar to the processing of reads in the VAMP-seq experiment, the tile reads were cleaned for adapter sequences using cutadapt[95] and paired-end reads were joined using fastq-join from ea-utils[97]. Only barcodes with an exact match to the barcode map were counted. If tiles from the Odds, Evens, or CT libraries were observed in a sorting of a different library, these were assumed to be non-sorted contaminants and ignored. Technical replicates of each FACS bin were merged and normalized to frequencies without pseudo counts. For each library,

biological, and FACS replicas, a tile stability index (TSI) was calculated per tile using:

$$\text{TSI}_t = \frac{\sum_g g \times f_{t,g}}{\sum_g f_{t,g}} \quad (5)$$

Where $f_{t,g}$ is the frequency of tile $t$ in FACS gate $g$. Two of the APPY-based control tiles, RLLL and DAAA, present in all sequenced pools were used to renormalize the Evens and CT libraries to match the TSI of the control tiles to the Odds library:

$$\text{TSI}_t^{\text{even,norm}} = 0.075 + 0.9018 * \text{TSI}_t^{\text{even}} \quad (6)$$

$$\text{TSI}_t^{\text{ct,norm}} = 0.5895 + 0.5570 * \text{TSI}_t^{\text{ct}} \quad (7)$$

Since the complexity of the libraries is relatively low, each tile is covered by more than 3500 observed reads per technical replicate on average. Thus 3 biological and 2 FACS replicates for each of the 3 libraries reproduced TSI scores very well with a minimum Pearson correlation of 0.97. The standard deviation over replicates is reported as an error estimate. The tile scores are available on the GitHub repository of this paper and the code is available with the original report of this experiment[76].

### Evolutionary conservation scores

Evolutionary distance from the WT sequence was calculated in silico for all the ASPA variants using information from evolutionary sequence conservation. We generated a multiple sequence alignment (MSA) of ASPA homologs using HHblits[98] with an E-value threshold of $10^{-20}$. The full ASPA MSA included 1102 sequences but was reduced to 757 homologs by filtering out sequences with more than 50% gaps. Using the MSA information, we calculated evolutionary conservation scores using the Global Epistatic Model for predicting Mutational Effects (GEMME) v1.0 software[54].

### In silico thermodynamic stability predictions

Changes in thermodynamic stability (ΔΔG) were predicted using Rosetta (GitHub SHA1 99d33ec59ce9fcecc5e4f3800c778a54afdf8504) with the Cartesian ddG protocol[53] on ASPA crystal structures 2O53, using only the chain A for the monomeric evaluation and both the chains (AB) for the dimeric evaluation. Non-protein atoms were removed from the crystal structure except for the zinc ion that was kept in the dimer calculations. All the ΔΔG values obtained from Rosetta were divided by 2.9 to bring them from Rosetta energy units onto a scale corresponding to kcal/mol[53] and truncated to the range 0–5 kcal/mol.

### RNA sequencing and RT-qPCR

After 5 days of induced protein expression using doxycycline, viable, singlet, mCherry-positive cells were sorted using a BD FACSJazz (BD Biosciences).

For the qPCR, RNA was purified from samples containing more than 500,000 cells using an RNeasy kit (Qiagen) following the protocol without the optional on-column DNase digestion step. DNA digestion was performed on 1 μg nucleic acid using DNase I, RNase-free (Thermo Fisher Scientific) as described by the manufacturer. 1 U/μL RiboLock RNase Inhibitor (Thermo Fisher Scientific), was included to prevent RNA degradation. Subsequently, reverse transcription was performed using Maxima H Minus Reverse Transcriptase (Thermo Fisher Scientific), yielding a 20 μL cDNA sample. Of these, 1 μL was mixed with 12.5 μL Maxima SYBR Green/ROX qPCR Master Mix (2X) (Thermo Fisher Scientific), primers at a final concentration of 0.3 μM for either HSPA1A/B or β-actin and water for a final volume of 25 μL. The samples were denatured at 95 °C (10 min), followed by 40 cycles performed at

the temperatures: 95 °C (15 s), 59 °C (30 s), and 72 °C (30 s). The mean of 3 replicates (not differing by more than 0.5 cycles) was calculated and normalized to β-actin. Subsequently, samples were normalized to a WT sample included in each PCR run. In total, 3 biological replicates were included per ASPA variant.

For the RNA sequencing, total RNA was isolated and purified from more than 1.5 million sorted cells using the GeneJET RNA Purification Kit (Thermo Fisher Scientific) according to the manufacturer's instructions. Genomic DNA was removed from 5 μg total RNA according to the manufacturer's instruction in the GeneJET RNA Purification Kit (Thermo Fisher Scientific). RNA samples of more than 2.5 μg were shipped to BGI (Hong Kong). The RNA was rRNA depleted and sequenced using DNBSEQ by BGI. The RNA sequencing was performed with three independent repeats (separate transfections) of each condition (WT and C152W).

### RNA sequencing data analysis

Quality control of sequence reads was done using the tools "FastQC" v0.11.7 (), "RSeQC" v2.6.4[99], and "fastq_screen" v0.11.4 (https://www.bioinformatics.babraham.ac.uk/projects/fastq_screen/). Low-quality bases and the first 12 bases and reads shorter than 25 nt were removed with "Trimmomatic" v0.39[100] using settings "HEADCROP:12 LEADING:3 SLIDINGWINDOW:4:15 MINLEN:35". Reads were mapped using "STAR" v2.7.3a[101] against the human genome (hg38). Up to two mismatches were allowed during the mapping, and the minimum number of overlap bases to trigger mates merging and realignment was set to five. Otherwise, default settings were used. Duplicate reads were also removed with the bamRemoveDuplicatesType "UniqueIdentical" option in "STAR". The "featureCounts" function of the "Rsubread" R package v2.2.6[102] was used to quantify reads in exons. The Gencode v38 comprehensive gene annotation including all genomic regions was used to assign reads to genes.

The "edgeR" v3.30.6 software[103] was used to perform a differential expression analysis. For this purpose, first, a model was defined indicating the experimental conditions. Library normalization factors were calculated using the "calcNormFactors" function with the "TMM" algorithm. Tag-wise dispersion was calculated using the "estimateDisp" function with "robust = TRUE". A gene-wise generalized linear model was fit with "glmQLFit". Finally, differential gene usage was assessed using "glmQLFTest". The resulting $p$-values were corrected for multiple testing using the "Benjamini–Hochberg" method.

### Statistics and reproducibility

Unless otherwise stated, all experiments were repeated independently at least three times.

### Reporting summary

Further information on research design is available in the Nature Portfolio Reporting Summary linked to this article.

## Data availability

The data generated in this study have been deposited on GitHub: https://github.com/KULL-Centre/_2023_Groenbaek-Thygesen_ASPA_MAVE (https://doi.org/10.5281/zenodo.8382504). The DNA sequencing data have been deposited at the Gene Expression Omnibus (GEO), accession code: GSE254639. Abundance and toxicity scores are also deposited at MaveDB (https://www.mavedb.org) under accession number urn:mavedb:00000657-a [https://www.mavedb.org/#/experiments/urn:mavedb:00000657-a]. Sequencing reads for the abundance and toxicity scores are available at https://doi.org/10.17894/ucph.3e05fe3a-4d7e-4d70-9056-18ed999e7e1e. The RNA seq. data have been uploaded to Gene Expression Omnibus (GEO): https://www.ncbi.nlm.nih.gov/geo/ (accession number: GSE232399; samples GSM7329952-57). The processed data are available in the source data file provided with this paper. Source data are provided with this paper.

## Code availability

The software used for this article is available at https://github.com/KULL-Centre/_2023_Groenbaek-Thygesen_ASPA_MAVE (https://doi.org/10.5281/zenodo.8382504).

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

## Acknowledgements

We acknowledge the use of the FACS, sequencing, microscopy, bioinformatics, and computing core facilities at the Biotech Research & Innovation Centre and the Department of Biology, University of Copenhagen. We thank Vibe H. Oestergaard, Michael Lisby, Leonor Rib, Nick Popp, Sofie V. Nielsen, Søren Lindemose, and Anne-Marie Lauridsen for technical support. Figures 1A, Fig. 5B, Supplementary Fig. 11C, created with BioRender.com, released under a Creative Commons Attribution-NonCommercial-NoDerivs 4.0 International license. The work was funded by the Novo Nordisk Foundation (https://novonordiskfonden.dk) challenge program PRISM (to K.L.-L., A.S., D.M.F. & R.H.-P.), the Novo Nordisk Foundation NNF21OC0071057 (to R.H.P.), the Lundbeck Foundation (https://www.lundbeckfonden.com) R272-2017-452 and R209-2015-3283 (to A.S.), and R249-2017-510 (to L.C.), and the Danish Council for Independent Research (Det Frie Forskningsråd) (https://dff.dk) https://doi.org/10.46540/2032-00007B (to R.H.P.).

## Author contributions

M.G.-T., V.V., K.E.J., M.C., L.P., T.K.S., L.C., S.N., R.L.P. and A.S. performed the experiments. M.G.-T., V.V., M.C., K.E.J., T.K.S., L.C., A.S., D.M.F., K.L.-L. and R.H.-P. analyzed the data. D.M.F., K.L.-L. and R.H.-P. conceived the study. M.G.-T., V.V. and R.H.-P. wrote the paper.

## Competing interests

K.L.-L. holds stock options in and is a consultant for Peptone Ltd. The remaining authors declare no competing interests.
