## [Peer Review File · Nature Communications]

Reviewers' Comments:

Reviewer #1:

Remarks to the Author:

The authors applied VAMP-seq to systematically characterize essentially all possible ASPA single-site missense and nonsense mutations in cultured cells, and then to map degradation signals (degrons) in these proteins. In this way, this manuscript generalizes findings by these authors employing a yeast expression system to characterize in detail a single mutated ASPA (ASPA C152W), which is known to be pathogenic in human Canavan disease, published in 2021 (Gersing et al, Nature Genetics 17(4):e1009539). In that prior paper, they suggested that this knowledge could be applied to identify potentially therapeutic modifiers of ASPA protein stability and loss of function.

The methods and data analyses are well described, and appear rigorous. Some discussion of the pluses and minuses of the cultured human cell system used in the present manuscript, vs the yeast system in their prior manuscript, would be appropriate.

As the authors point out, Canavan disease is a recessive, and hence the knowledge gained by this approach is less apt to be directly therapeutically applicable than when a dominant negative mechanism is in play. Thus, although the authors state that such detailed studies "are essential steps toward future implementation of precision medicine for Canavan's disease", the road toward such implementation is far from clear. This somewhat weakens the significance of the present study, though the authors do point to cystic fibrosis as an example in which drugs that can reactivate pathogenic variants have been developed. The authors also state that the "comprehensive genotype-phenotype information" they present will "provide an essential step toward a future implementation of gene therapy or precision medicine approaches for Canavan's disease". One could agree that this would be the case for genetic diseases in which early markers of pathogenicity are not available, but this would not be the case for Canavan's disease, in which elevations in N-acetyl-L-aspartate are readily demonstrable prior to the onset of overt neurodegeneration.

It would have been desirable to provide at least some context concerning other known pathogenic ASPA mutations in the present manuscript. For example, several of these mutations have been associated with some retained ASPA enzymic activity and with juvenile, rather than infantile, disease onset.

Despite the above caveats, the approach that the authors have taken in the present manuscript can provide "comprehensive genotype-phenotype information", and hence should be applicable to a variety of diseases resulting from missense or nonsense mutations.

Reviewer #2:

Remarks to the Author:

This study explores the relationship between sequence variation and protein stability by employing human aspartoacylase (ASPA) as a model. The manuscript serves as a follow-up to the authors' earlier work, which examined ASPA C152W, a loss-of-function variant linked to Canavan disease, in the context of cellular protein quality control (PQC) pathways. In the present work, the authors performed systematic studies on ASPA variants, covering virtually all 6000+ possible single-site mutations and assessing their abundance in HEK293T cells. Protein abundance was measured using fluorescence-activated cell sorting coupled with Variant Abundance by Massively Parallel sequencing. The approach enabled thorough investigation of ASPA variants, correlating their impact on predicted thermodynamic stability, evolutionary conservation, and cellular fitness.

The study constitutes a comprehensive investigation of destabilized variants of a singular protein, offering value to the field. Deep mutational scanning unveiled key regions in the ASPA protein crucial for its stability within cells. Intriguingly, it also identified outlier groups of missense mutations that deviate from the predicted effects. While the work represents a remarkable effort, the robustness of the conclusions could be strengthened in the following ways:

1) The authors previously reported that wildtype ASPA forms foci in yeast and is partially insoluble when expressed in human cells (Gersing et al. 2021 Plos Genet 17:e1009539). It is crucial that the properties of the ASPA protein is thoroughly characterized in the HEK293T expression system, both using fluorescence microscopy and biochemical means (including fractionation experiments), as the potential presence of a misfolded or aggregated fraction complicates the interpretation of the results. I recommend performing this characterization on both the WT and variants with a wide range of different stabilities. This analysis establishes the foundation for correlating abundance measurements to protein stability.

2) The examination of the effects of temperature and chemical perturbations, as currently depicted in Fig. 4, contributes to demonstrating the dynamic range of the experimental system, and as such, should be added to this characterization. Additionally, these datasets need to be supported by experiments with WT and known variants in all conditions to enable the interpretation of the data obtained with the ASPA library.

3) While the present study examines the impact of missense mutations on protein stability inside cells, it does not explicitly address the direct relationship with PCQ components, nor does it delve into their potential collaboration. Therefore, the authors' claim that the study provides mechanistic insight into the targeting of ASPA variants by PQC systems is not supported by the data. This should be revised throughout in the text.

4) The results of this study heavily depend on the capability to isolate individual cells from a mixed population. However, the methodology employed to accomplish this objective is inadequately described in the Materials and Methods section. Considering that the diameter of Hek293 cells ranges from 10-20 microns, using a 50-micron mesh for filtration is insufficient for achieving a single-cell suspension. What precautions did the authors take to detach the cells from each other and from the surface (assuming that they were grown as an adherent monolayer)? How did the authors ensure that the cells were not sticking to each other? What FACS parameters were used to identify the cell population? Additionally, the gating strategy used for cell sorting should be provided as a supplemental figure.

5) Based on the methodology, it appears to me that the "toxicity screen" is in fact a competition assay. Unless the authors have observed a clear phenotype that warrants the use of the term "toxicity" in the context of the ASPA variants identified in this assay, such as reduced cell viability or markers of cell death, I recommend replacing this word with "fitness" throughout the text.

6) The outlier groups of missense mutations that deviate from the predicted effects are interesting and should be further investigated and discussed. In fact, the presence of outliers offers room for speculation with respect to PQC engagement inside cells. I would like to see a 3D representation of outlier positions in the ASPA structure, as well as biochemical data and fluorescence micrographs to show the subcellular distribution for representative groups.

7) Considering ASPA is an enzyme, an obvious question arises: how do the missense mutations, or representative groups thereof, impact its enzymatic activity? Employing a functional readout could aid in annotating the different variants – especially the variants of uncertain significance in Figure 6. Is it feasible for the authors to investigate this aspect with representative groups of variants?

8) Beyond influencing conformational stability, missense mutations can lead to various other outcomes. They might introduce new sites that are targeted by proteases or posttranslational modifications, or create potential interfaces for protein interactions such as HSP70. As a result, these effects might be responsible for unexpected behavior of ASPA variants. This perspective should be explored.

9) Given that the approach provides a comprehensive view of protein structure in various contexts, I recommend consolidating all this information into a final 3D model illustrating structural "hotspots" for protein stability.

10) In Figures 2a, 7a, S5, S6, it would be helpful to indicate regions in the protein that are critical

for enzymatic activity, such as those that form the active site and in contact with the Zn²⁺ ion. I also suggest indicating the "Missing variants" in black color as opposed to grey for better visibility on printed paper.

11) The Methods section should clarify when temperature treatment of chemical perturbants were applied to the cells. Right now, it is unclear whether the treatments happened before or after harvesting of cells.

Reviewer #3:

Remarks to the Author:

This is an exciting paper looking at the effects of mutations of the human enzyme called aspartoacylase (ASPA). Loss-of-function of this enzyme is linked to Canavan's disease, an autosomal recessive and lethal neurological disorder. The authors do not focus on understanding how mutations alter the activity of the enzyme but instead focus on mutations that affect its stability, and thus likely its abundance, and also its toxicity, as misfolded protein sometimes can take part in uncontrolled, unwanted interactions with other proteins, which leads to toxicity. Overall, the data appears to have been produced rigorously, and data interpretation is done with the support of additional validation experiments and rigorous computational analyses. I noted a few elements, some general, some specific, that could help improve the manuscript. They are listed below.

1) Generally, it would have been interesting to know if this protein represents a protein family or if its domain has not so far been investigated by deep mutational scanning so the results can have a broader impact than variant interpretation for the disease itself.

2) It would have been interesting to discuss further that the enzyme is active as a monomer, yet it also forms a homodimer with a large interface. This is an autosomal recessive disease, so a single WT copy of the enzyme provides enough activity. Still, the toxicity uncovered here suggests that some variants could also be dominant. Are there reported cases? In the analysis of gnomAD and polymorphism data, is there a further decrease or depletion in allele frequency for variants that have the potential to be partially dominant?

3) I understand that the WT copy of the enzyme is not detectable in the cell lines, so its presence should not interfere with the experiments performed. However, if the enzyme is not expressed in the cells, does it mean the impact we see for some variants in this cell line is irrelevant? Suppose the physiology of these cells is not "tuned" to processing this protein. Does it mean that the interactions with the degradation and chaperone system uncovered are not what would generally happen for these variants in cells where this enzyme is commonly expressed?

4) In general, it would be necessary to discuss better the mutations that likely impact the enzyme's activity and how these mutations differ from the destabilizing ones. From reading the manuscript, we have the impression that all that matters is stability (which could be true). This is likely essential, but are relatively more disease mutations acting through this mechanism than expected? Is there a way to also model the interaction with the substrate so this effect can be considered? There are some mentions here and there of this question, but it would be interesting to treat it more explicitly.

5) I was surprised to see that cells were sorted in only four bins. Is this what is routinely done for these types of assays? Is this assay limited to that, or was this optimal?

6) The "however" in line 203 may not be the most appropriate word here.

7) Line 208: Briefly mention what WCN measures and what it represents for a protein.

8) Line 254-255: would it be beneficial for the reader to show this for a few examples, for instance, catalytically essential residues?

9) For Figure 4, I was wondering if the various processes probed here were occurring at the same time scale, for instance, protein degradation by the proteasome versus by autophagy, and if the experiments were designed to accommodate these differences so the roles of some of them can actually be observed with the assays and at the level of expression used.

10) Figure 7B, C,D: it would be better to use another colour scale so as not to use green, which is used to represent other things in the other panels. Also, would the abundance score in B be better illustrated on a log scale?

11) Line 395, 396: I need clarification on this statement. To me, this means that toxic variants will

be rare in the experiment, and this would be supported by low read count and thus provide limited resolution on their effects as seen from abundance measurement. This needs to be clarified. This is not a significant issue because this is the protein's biology, but this must be discussed and controlled clearly.

12) It would have been helpful to do some microscopy on some destabilized toxic proteins and some that are not toxic to see if they behave differently regarding aggregation, for instance.

13) Line 436: there is no file called (SupplementalFile4.xlsx)

14) I was trying to understand the statement on lines 595-597. This could potentially lead to erroneous results or noise in the data. What was the count difference between the variants that mapped to the same barcode when this was done? If the frequencies are sometimes similar, it is likely incorrect to do that.

15) How many replicates were performed for the RNAseq analyses needed to be made clear.

16) In general, the figure legends of the supp. material is much less detailed than the main figures. Figure S4 is a good example. All legends should be expanded and more precise.

17) No colour scale is shown in Figure S3, so we do not know how to interpret the colours.

18) For Figures S5 and S6, why not show scatter plots also so we see the covariance between the various measures? Or are those in the main text? Maybe mention in the legend of those if the data is also shown in the main text.

19) Figure S10: many non-sense mutants are also toxic. Do their positions fit with the degran mapping data?

20) Figure S11: Resolution appears to be lower quality than other figures.

Our point-by-point response

Reviewer #1:

The authors applied VAMP-seq to systematically characterize essentially all possible ASPA single-site missense and nonsense mutations in cultured cells, and then to map degradation signals (degrons) in these proteins. In this way, this manuscript generalizes findings by these authors employing a yeast expression system to characterize in detail a single mutated ASPA (ASPA C152W), which is known to be pathogenic in human Canavan disease, published in 2021 (Gersing et al, Nature Genetics 17(4):e1009539). In that prior paper, they suggested that this knowledge could be applied to identify potentially therapeutic modifiers of ASPA protein stability and loss of function.

The methods and data analyses are well described, and appear rigorous. Some discussion of the pluses and minuses of the cultured human cell system used in the present manuscript, vs the yeast system in their prior manuscript, would be appropriate.

Our response:

We thank the reviewer for the kind words on our work and for spending time to provide valuable feedback.

In the Discussions section of the revised manuscript we now include some remarks on the differences and similarities and pros/cons of using cultured human cells vs. yeast (p.19).

As the authors point out, Canavan disease is a recessive, and hence the knowledge gained by this approach is less apt to be directly therapeutically applicable than when a dominant negative mechanism is in play. Thus, although the authors state that such detailed studies "are essential steps toward future implementation of precision medicine for Canavan's disease", the road toward such implementation is far from clear. This somewhat weakens the significance of the present study, though the authors do point to cystic fibrosis as an example in which drugs that can reactivate pathogenic variants have been developed. The authors also state that the "comprehensive genotype-phenotype information" they present will "provide an essential step toward a future implementation of gene therapy or precision medicine approaches for Canavan's disease". One could agree that this would be the case for genetic diseases in which early markers of pathogenicity are not available, but this would not be the case for Canavan's disease, in which elevations in N-acetyl-L-aspartate are readily demonstrable prior to the onset of overt neurodegeneration.

Our response:

We agree and have toned down the remarks on the potential clinical use of our data (p.22).

It would have been desirable to provide at least some context concerning other known pathogenic ASPA mutations in the present manuscript. For example, several of these mutations have been associated with some retained ASPA enzymic activity and with juvenile, rather than infantile, disease onset.

Our response:

We thank the reviewer for bringing this to our attention, and we now include a brief section on this (p.14-15).

Despite the above caveats, the approach that the authors have taken in the present manuscript can provide "comprehensive genotype-phenotype information", and hence should be applicable to a variety of diseases resulting from missense or nonsense mutations.

Our response:
Thank you.

Reviewer #2:

This study explores the relationship between sequence variation and protein stability by employing human aspartoacylase (ASPA) as a model. The manuscript serves as a follow-up to the authors' earlier work, which examined ASPA C152W, a loss-of-function variant linked to Canavan disease, in the context of cellular protein quality control (PQC) pathways. In the present work, the authors performed systematic studies on ASPA variants, covering virtually all 6000+ possible single-site mutations and assessing their abundance in HEK293T cells. Protein abundance was measured using fluorescence-activated cell sorting coupled with Variant Abundance by Massively Parallel sequencing. The approach enabled thorough investigation of ASPA variants, correlating their impact on predicted thermodynamic stability, evolutionary conservation, and cellular fitness.

The study constitutes a comprehensive investigation of destabilized variants of a singular protein, offering value to the field. Deep mutational scanning unveiled key regions in the ASPA protein crucial for its stability within cells. Intriguingly, it also identified outlier groups of missense mutations that deviate from the predicted effects. While the work represents a remarkable effort, the robustness of the conclusions could be strengthened in the following ways:

- 1) The authors previously reported that wildtype ASPA forms foci in yeast and is partially insoluble when expressed in human cells (Gersing et al. 2021 Plos Genet 17:e1009539). It is crucial that the properties of the ASPA protein is thoroughly characterized in the HEK293T expression system, both using fluorescence microscopy and biochemical means (including fractionation experiments), as the potential presence of a misfolded or aggregated fraction complicates the interpretation of the results. I recommend performing this characterization on both the WT and variants with a wide range of different stabilities. This analysis establishes the foundation for correlating abundance measurements to protein stability.

Our response:

We thank the reviewer for spending time to provide valuable feedback on our work.

Indeed, we previously noted aggregation of both C152W and, to a lesser extent, WT ASPA when expressed in yeast cells. We believe this was caused by overexpression in that system. Using the present expression system, where the ASPA variants are expressed from a single copy (integrated in the HEK293T genome), it is unlikely that we will achieve as high levels as was possible by episomal expression in yeast cells. Accordingly, we do not observe aggregates of neither WT nor C152W with the current system (Fig. 1B). In the revised manuscript, we expanded these analyses to include additional ASPA variants. None of these form visible aggregates (new Supplementary Fig. 3A). We also compared the solubility of these variants by differential centrifugation (new Supplementary Fig. 3B). Indeed, the amount of soluble C152W is strongly reduced, but in all cases, including WT ASPA, a small amount of insoluble protein is present. Based on these experiments we find that any potential aggregation with the landing-pad-based expression system is minor and certainly much lower than that we observed in yeast. Effects of potential aggregation are therefore small and should not affect

the interpretation of our high-throughput screen. These experiments thus help us establish the biochemical background for interpreting the VAMP-seq data in terms of abundance (p.8).

2) The examination of the effects of temperature and chemical perturbations, as currently depicted in Fig. 4, contributes to demonstrating the dynamic range of the experimental system, and as such, should be added to this characterization. Additionally, these datasets need to be supported by experiments with WT and known variants in all conditions to enable the interpretation of the data obtained with the ASPA library.

Our response:

The purpose of the results presented in Fig. 4 is to reveal (on a global scale) how the distribution of the library changes in response to various conditions that may affect the cellular protein quality control system. Since we do not sort and sequence the cells after the perturbations, the changes in the distributions caused by the perturbations (Fig. 4) do not quantify the dynamic range of the VAMP-seq assay. Instead, the dynamic range of the VAMP-seq assay is shown in Fig. 1E and is about 10-fold.

As additional controls for the experiments in Fig. 4, we have now performed flow cytometry on WT and C152W under the eight different conditions shown in Fig. 4. These data are included as a new figure (Supplemental Fig. 8). The results show, as expected, that WT is largely unresponsive to the perturbations, while the abundance of the C152W increases in response to reduced temperature, MLN7243, bortezomib, and YM01. The abundance of C152W is reduced at an elevated temperature, and is unchanged by chloroquine and NAA. The reason why bortezomib treatment, for instance, does not result in the C152W abundance increasing to the same level as untreated WT, is likely due to the limited time of the treatment. Although we did not test it directly, we are worried that longer treatments will kill the cells. Based on the new data, we argue that our interpretation of the library experiments (Fig. 4) is robust.

3) While the present study examines the impact of missense mutations on protein stability inside cells, it does not explicitly address the direct relationship with PCQ components, nor does it delve into their potential collaboration. Therefore, the authors' claim that the study provides mechanistic insight into the targeting of ASPA variants by PQC systems is not supported by the data. This should be revised throughout in the text.

Our response:

We apologize for this and have changed the text accordingly throughout (e.g. p.2).

4) The results of this study heavily depend on the capability to isolate individual cells from a mixed population. However, the methodology employed to accomplish this objective is inadequately described in the Materials and Methods section. Considering that the diameter of Hek293 cells ranges from 10-20 microns, using a 50-micron mesh for filtration is insufficient for achieving a single-cell suspension. What precautions did the authors take to detach the cells from each other and from the surface (assuming that they were grown as an adherent monolayer)? How did the authors ensure that the cells were not sticking to each other? What FACS parameters were used to identify the cell population? Additionally, the gating strategy used for cell sorting should be provided as a supplemental figure.

Our response:

We apologize for not including this information. In the revised manuscript, we depict the employed gating strategy (Supplemental Fig. 20), which includes a selection for single cells. In addition, we stress that we apply extensive trypsination prior to sorting (p.29).

5) Based on the methodology, it appears to me that the “toxicity screen” is in fact a competition assay. Unless the authors have observed a clear phenotype that warrants the use of the term “toxicity” in the context of the ASPA variants identified in this assay, such as reduced cell viability or markers of cell death, I recommend replacing this word with “fitness” throughout the text.

Our response:

We thank the reviewer for pointing this out. Indeed, the growth assay is based on competition between the different ASPA variants. However, as we show directly for the C152W variant, the reduced “fitness” is very severe, with expression of these variants causing a very substantial drop in growth; fitness or reduced fitness, therefore seems to us too mild a term to properly convey the situation. In competition-based assays, small effects can sometimes become significant when the cells are cultured over long periods of time, but our studies of individual variants show that at least some of the most “toxic” variants are rapidly lost from the population. Toxicity, similar to fitness, is unlike cell proliferation, cell viability, and cell death, not a clearly biologically defined term. We therefore deliberately chose this term, and the resulting “toxicity score” over a potential “fitness score”. For these reasons, we would prefer to stick with this nomenclature. However, to avoid potentially confusing readers of our paper, we have introduced a paragraph in the results section (p.15) to make this clear.

6) The outlier groups of missense mutations that deviate from the predicted effects are interesting and should be further investigated and discussed. In fact, the presence of outliers offers room for speculation with respect to PQC engagement inside cells. I would like to see a 3D representation of outlier positions in the ASPA structure, as well as biochemical data and fluorescence micrographs to show the subcellular distribution for representative groups.

Our response:

While we fundamentally agree that analyzing outliers is a useful approach to gain additional understanding, we also note that the outliers can be the result of imprecise experimental data and/or because the in silico prediction tools are imperfect. Indeed, both GEMME and Rosetta have limitations and have been noted to be imperfect (PMID: 37468638; PMID: 37563330). In addition, since neither GEMME nor Rosetta are predictors of protein abundance, it is unclear exactly which sort of correlation (linear, sigmoidal, etc.) is expected with protein abundance. Moreover, for GEMME, residues can be conserved for functional purposes that are independent of the protein structure (e.g. an exposed motif or binding site). On the other hand, variants that generate a degron may be degraded without destabilizing the native fold and would therefore not be detected by Rosetta. Accordingly, we feel that delving deeper into these outliers is beyond the scope of the current paper.

We have expanded the discussion with some of these points and note that as more and more proteins are analyzed by deep mutational scanning, it may be possible to develop more accurate computational predictors of protein abundance (p.21).

7) Considering ASPA is an enzyme, an obvious question arises: how do the missense mutations, or representative groups thereof, impact its enzymatic activity? Employing a functional readout could aid in annotating the different variants – especially the variants of uncertain significance in Figure 6. Is it feasible for the authors to investigate this aspect with representative groups of variants?

Our response:

We agree that this would be a nice addition to the paper. Unfortunately, the current methods for assaying ASPA activity are not straightforward and generally require purification of the proteins, which is likely to be particularly troublesome for low abundance variants. Previously, we attempted to establish a yeast-based activity assay for ASPA function (Gersing et al. 2021, PMID: 33914734). Unfortunately, as we reported, this was unsuccessful. For these reasons, we have not analyzed the enzyme activity of the ASPA variants. In the discussion (p.22), we now mention that activity-based screening of our ASPA library would be interesting if a scalable cell-based system is established.

8) Beyond influencing conformational stability, missense mutations can lead to various other outcomes. They might introduce new sites that are targeted by proteases or posttranslational modifications, or create potential interfaces for protein interactions such as HSP70. As a result, these effects might be responsible for unexpected behavior of ASPA variants. This perspective should be explored.

Our response:

We agree that indeed this is possible. Accordingly, we now mention this in the discussion (p.21). In addition, we would like to point out again that Rosetta is an imperfect predictor of changes in thermodynamic protein stability and it is therefore currently unclear exactly which variants behave unexpectedly.

9) Given that the approach provides a comprehensive view of protein structure in various contexts, I recommend consolidating all this information into a final 3D model illustrating structural “hotspots” for protein stability.

Our response:

We are not sure what kind of visualization the reviewer is asking for, but note that we already provide a protein structure colour-coded using the VAMP-seq scores (Fig. 2D) and toxicity scores (Fig. 7E).

10) In Figures 2a, 7a, S5, S6, it would be helpful to indicate regions in the protein that are critical for enzymatic activity, such as those that form the active site and in contact with the Zn²⁺ ion. I also suggest indicating the “Missing variants” in black color as opposed to grey for better visibility on printed paper.

Our response:

We agree that highlighting residues critical for activity is a good idea and now provide this (represented by the black bar) below the heatmaps (Fig. 2A, supplementary Fig. 6 and 7). Since grey is commonly used to denote missing variants in deep mutational scans, we would prefer not to change the coloring of the missing variants in the heatmaps.

11) The Methods section should clarify when temperature treatment of chemical perturbants were applied to the cells. Right now, it is unclear whether the treatments happened before or after harvesting of cells.

Our response:

We thank the reviewer for pointing this out. We have attempted to make this clearer in the methods (p.29) section and the figure legends.

Reviewer #3 (Remarks to the Author):

This is an exciting paper looking at the effects of mutations of the human enzyme called aspartoacylase (ASPA). Loss-of-function of this enzyme is linked to Canavan's disease, an autosomal recessive and lethal neurological disorder. The authors do not focus on understanding how mutations alter the activity of the enzyme but instead focus on mutations that affect its stability, and thus likely its abundance, and also its toxicity, as misfolded protein sometimes can take part in uncontrolled, unwanted interactions with other proteins, which leads to toxicity. Overall, the data appears to have been produced rigorously, and data interpretation is done with the support of additional validation experiments and rigorous computational analyses. I noted a few elements, some general, some specific, that could help improve the manuscript. They are listed below.

Our response:

We thank the reviewer for the kind words and for spending time to provide valuable feedback on our work.

1) Generally, it would have been interesting to know if this protein represents a protein family or if its domain has not so far been investigated by deep mutational scanning so the results can have a broader impact than variant interpretation for the disease itself.

Our response:

This is a good point. ASPA displays some structural similarity to Zn²⁺-dependent carboxypeptidase A-related hydrolases. As far as we are aware, none of the related proteins have been analyzed by deep mutational scanning. We now mention this in the manuscript (p.3-4).

2) It would have been interesting to discuss further that the enzyme is active as a monomer, yet it also forms a homodimer with a large interface. This is an autosomal recessive disease, so a single WT copy of the enzyme provides enough activity. Still, the toxicity uncovered here suggests that some variants could also be dominant. Are there reported cases? In the analysis of gnomAD and polymorphism data, is there a further decrease or depletion in allele frequency for variants that have the potential to be partially dominant?

Our response:

We agree and elaborate on this in the results (p.17) and discussion (p.19). To the best of our knowledge, there are no indications that Canavan disease can be dominant. Accordingly, the reduced growth we observe upon expression of many low abundance variants we believe to be a consequence of them being overexpressed. Thus, although this may not be medically relevant, it reveals on a large

scale that destabilized protein variants can result in growth defects which on its own is interesting and may be relevant for other implementations of the VAMP seq. technique. We have included a plot of the toxicity vs. gnomAD allele frequency (Supplementary Fig. 18). This shows, as expected, that the toxic variants are rare. To examine this issue in more detail, we followed the reviewer's suggestion and compared the gnomAD allele frequencies of different groups of variants. Specifically, we divided variants into three groups: (i) low abundance and low toxicity, (ii) low abundance and high toxicity and (iii) high abundance (these all have low toxicity). We find that the average and median allele frequency in each class is comparable (within error), though all the most common variants (with allele frequencies above 10^{-3}) have high abundance (and low toxicity). Thus, our data do not show evidence of a depletion of toxic (class ii) variants compared to non-toxic low-abundance variants (class i). These results are briefly discussed in the revised paper (p.17).

3) I understand that the WT copy of the enzyme is not detectable in the cell lines, so its presence should not interfere with the experiments performed. However, if the enzyme is not expressed in the cells, does it mean the impact we see for some variants in this cell line is irrelevant? Suppose the physiology of these cells is not "tuned" to processing this protein. Does it mean that the interactions with the degradation and chaperone system uncovered are not what would generally happen for these variants in cells where this enzyme is commonly expressed?

Our response:

The role of ASPA in Canavan disease appears to be particular to the oligodendrocytes in the brain, though the gene is widely expressed and particularly highly expressed in kidney (PMID: 22750302). Although, HEK293T cells are derived from kidney we are unable to detect expression of ASPA in the cells. We agree that this may be a concern and now mention this as a potential caveat in the manuscript (p.19). However, we also note that the cellular protein quality system is broadly expressed in human cells and highly conserved through evolution. Hence, likely the general mechanism of PQC in oligodendrocytes is similar to that in HEK293T cells.

4) In general, it would be necessary to discuss better the mutations that likely impact the enzyme's activity and how these mutations differ from the destabilizing ones. From reading the manuscript, we have the impression that all that matters is stability (which could be true). This is likely essential, but are relatively more disease mutations acting through this mechanism than expected? Is there a way to also model the interaction with the substrate so this effect can be considered? There are some mentions here and there of this question, but it would be interesting to treat it more explicitly.

Our response:

This is related to point 10 of Reviewer #2. We agree and have marked the positions (black bars) in the heatmaps of the residues that engage in the catalytic activity of the enzyme (Fig. 2A and supplementary Fig. 6 & 7). Previously, we have used combined analyses of evolution and protein structure to disentangle residues that are important for structural stability vs those that are importance e.g. for catalysis (PMID: 37443362). In the current manuscript we performed similar analysis and indeed find a small number of pathogenic ASPA variants that appear to be important for function (as assessed by evolutionary conservation) but not abundance/stability (Fig. 6A). In addition, we include the structure of ASPA marked with the catalytic sites as well as site predicted to be unimportant for stability, but important for function (supplementary Fig. 10). We discuss this point in the revised manuscript on p.14.

5) I was surprised to see that cells were sorted in only four bins. Is this what is routinely done for these types of assays? Is this assay limited to that, or was this optimal?

Our response:

Previous VAMP seq. analyses have also relied on sorting into four bins (PMID: 29785012). In retrospect (considering the toxicity of the low abundance variants) it would have been preferable to have screened with more bins. We now mention this in the manuscript (p.21).

6) The “however” in line 203 may not be the most appropriate word here.

Our response:

Thank you. We have corrected this.

7) Line 208: Briefly mention what WCN measures and what it represents for a protein.

Our response:

WCN is the weighted contact number and reports on how many contacts a residue forms, and is thus also related to burial/solvent exposure. We now include this information (p.9).

8) Line 254-255: would it be beneficial for the reader to show this for a few examples, for instance, catalytically essential residues?

Our response:

Thank you for pointing this out. We hope this is now clearer in the manuscript and Supplementary Fig. 10 (please see also our response to point 4 above).

9) For Figure 4, I was wondering if the various processes probed here were occurring at the same time scale, for instance, protein degradation by the proteasome versus by autophagy, and if the experiments were designed to accommodate these differences so the roles of some of them can actually be observed with the assays and at the level of expression used.

Our response:

Indeed, this is possible. Due to the observed toxicity of ASPA, as well as the potential toxic effects of longer treatments and the stabilities of the drugs, we are limited to fairly short timescales. We modified the manuscript, so the Fig 4 legend now includes information for how long the cells were treated. In addition, we mention this point in the revised manuscript (p.11 & 12). Finally, we note that our analyses of WT and C152W (new supplementary Fig. 8) are in line with our interpretation of the results in Fig. 4.

10) Figure 7B, C,D: it would be better to use another colour scale so as not to use green, which is used to represent other things in the other panels. Also, would the abundance score in B be better illustrated on a log scale?

Our response:

Thank you. We have changed the color scale in Fig. 7BCD (and to match Fig. 7, also supplementary Fig. 16). Since the abundance score of the toxic variants is in all cases essentially 0 or even slightly negative, we did not change the axis to a log scale.

11) Line 395, 396: I need clarification on this statement. To me, this means that toxic variants will be rare in the experiment, and this would be supported by low read count and thus provide limited resolution on their effects as seen from abundance measurement. This needs to be clarified. This is not a significant issue because this is the protein's biology, but this must be discussed and controlled clearly.

Our response:

Thanks for pointing out that this text is indeed not very clear. Firstly, toxicity is measured as a decrease in read counts relative to wild type and we have sufficient sequencing depth to have nicely reproducible toxicity scores also for the more toxic variants (Supplementary Fig. 12), i.e. there are still sufficient read counts despite the decrease. The mentioned resolution refers to the FACS-based abundance screen where the low-abundance FACS gate is wide (Fig. 1F), resulting in abundance score of zero covering a wide range of fluorescence intensities (GFP:mCherry). This is confirmed in Fig. 2C where a range of low-throughput fluorescence measurements (0.2-0.8) maps to a high-throughput abundance score of approximately zero. It is this peak of zero-abundance-score variants that may be further resolved by the toxicity score because this, as indicated in Supplementary Fig. 14, may reflect cellular abundance levels with a different dynamic range. We now clarify this in the manuscript (p.16).

12) It would have been helpful to do some microscopy on some destabilized toxic proteins and some that are not toxic to see if they behave differently regarding aggregation, for instance.

Our response:

This point was also made by Reviewer #2 (point 1).

Indeed, we previously noted aggregation of both C152W and, to a lesser extent, WT ASPA when expressed in yeast cells. We believe this was caused by substantial overexpression in yeast. Using the present expression system, where the ASPA variants are expressed from a single copy (integrated in the HEK293T genome), it is unlikely that we achieve as high levels as in episomal expression in yeast. Accordingly, we do not observe aggregates of either WT or C152W with the current system (Fig. 1B). In the revised manuscript, we expanded these analyses to include additional ASPA variants (Supplementary Fig. 3A). None of these form notable aggregates. We also compared the solubility of these variants by differential centrifugation (Supplementary Fig. 3B). Indeed, the amount of soluble C152W is reduced, but in all cases, including WT ASPA, some insoluble protein is present. Hence, for these ASPA variants, aggregation does not seem relevant for our interpretation of the screening data (p.8).

13) Line 436: there is no file called (SupplementalFile4.xlsx)

Our response:

We apologize. This data is now incorporated into one supplemental excel spreadsheet (Source Data).

14) I was trying to understand the statement on lines 595-597. This could potentially lead to erroneous results or noise in the data. What was the count difference between the variants that mapped to the same barcode when this was done? If the frequencies are sometimes similar, it is likely incorrect to do that.

Our response:

Out of 134,740 barcodes, 1,436 (1%) mapped to more than one ASPA DNA variant with 1,425 mapping to two variants and 11 mapping to more. We expect this mainly to be due to (possibly biased) sequencing errors. Typically, one ASPA barcode had a majority of reads, on average 89% of reads, and only 29 barcodes had less than 60% of reads assigned to a dominant variant. Also, the described filtering of PacBio reads heavily reduces this problem, suggesting sequencing errors as the source; thus, some of the 29 barcodes may represent noise. We have included these details in the text (p.26) to justify the approach taken.

15) How many replicates were performed for the RNAseq analyses needed to be made clear.

Our response:

The RNA sequencing was performed with three independent repeats (separate transfections) of each condition (WT and C152W). We now state this in the figure legend (supplementary Fig. 19) and methods section (p.38).

16) In general, the figure legends of the supp. material is much less detailed than the main figures. Figure S4 is a good example. All legends should be expanded and more precise.

Our response:

We have expanded several of the supplemental figure legends.

17) No colour scale is shown in Figure S3, so we do not know how to interpret the colours.

Our response:

Thank you. In the revision a color scale is now included (supplementary Fig. 4).

18) For Figures S5 and S6, why not show scatter plots also so we see the covariance between the various measures? Or are those in the main text? Maybe mention in the legend of those if the data is also shown in the main text.

Our response:

Thank you for pointing this out. Indeed, the scatter plots are shown in the main manuscript (Fig. 3). We now mention this in the legends of the supporting figures.

19) Figure S10: many non-sense mutants are also toxic. Do their positions fit with the degron mapping data?

Our response:

This is an interesting idea. We examined this and did not find any clear correlation between the location of the toxic non-sense variants and the degrons, except that (as we already noted in the manuscript) many toxic nonsense variants appear to cluster towards the C-terminus of ASPA (where the bulk of ASPA has been produced). We now mention (p.20) these are also ASPA variants that include many degrons.

20) Figure S11: Resolution appears to be lower quality than other figures.

Our response:

Thank you. The revision includes a figure with higher resolution (Supplementary Fig. 14).

Reviewers' Comments:

Reviewer #2:

Remarks to the Author:

The authors have addressed the questions/issues raised in my initial review and I am satisfied with their revisions. The work constitutes a remarkable effort that will be of value to the field. I recommend that the revised manuscript be accepted for publication in Nature Communications.

Reviewer #3:

Remarks to the Author:

The responses to my comments are satisfying. Thanks to the authors for their additional work.

OUR RESPONSE TO THE REVIEWER COMMENTS

Reviewer #2 (Remarks to the Author):

The authors have addressed the questions/issues raised in my initial review and I am satisfied with their revisions. The work constitutes a remarkable effort that will be of value to the field. I recommend that the revised manuscript be accepted for publication in Nature Communications.

Reviewer #3 (Remarks to the Author):

The responses to my comments are satisfying. Thanks to the authors for their additional work.

Our response:

We thank the reviewers for their time and positive evaluation of our work.